# Insights into the mechanism of action of the arbitrium communication system in SPbeta phages

Francisca Gallego del Sol [1], Nuria Quiles-Puchalt[2], Aisling Brady [2,3], José R. Penadés [2✉] & Alberto Marina [1✉]

The arbitrium system is employed by phages of the SPbeta family to communicate with their progeny during infection to decide either to follow the lytic or the lysogenic cycle. The system is controlled by a peptide, AimP, that binds to the regulator AimR, inhibiting its DNA-binding activity and expression of *aim*X. Although the structure of AimR has been elucidated for phages SPβ and phi3T, there is still controversy regarding the molecular mechanism of AimR function, with two different proposed models for SPβ. In this study, we deepen our understanding of the system by solving the structure of an additional AimR that shows chimerical characteristics with the SPβ receptor. The crystal structures of this AimR (apo, AimP-bound and DNA-bound) together with in vitro and in vivo analyses confirm a mechanism of action by AimP-induced conformational restriction, shedding light on peptide specificity and cross regulation with relevant biological implications.

[1] Instituto de Biomedicina de Valencia (IBV), CSIC and CIBER de Enfermedades Raras (CIBERER), Valencia, Spain. [2] MRC Centre for Molecular Bacteriology and Infection, Imperial College London, London, UK. [3] Institute of Infection, Immunity and Inflammation, College of Medical, Veterinary and Life Sciences, University of Glasgow, Glasgow, UK. ✉email: j.penades@imperial.ac.uk; amarina@ibv.csic.es

Communication between different members of a community is not restricted to highly evolved animals but is also common in microorganisms that use quorum-sensing (QS) mechanisms for cell-cell information exchange. In Gram-positive bacteria, QS is mostly dominated by the synthesis and secretion of small peptides that work as signalling molecules monitoring population density[1]. These peptides are sensed in the medium by membrane-bound histidine kinases from two component systems, or in the bacterial cytoplasm by regulatory receptors after active importation. The RRNPP family (named after the representative constituent subfamilies Rgg, Rap, Npr, PlcR and PrgX) is the largest QS family of cytoplasmic regulatory receptors in Firmicutes. RRNPP receptors regulate a wide array of key biological processes including virulence, sporulation, biofilm formation and natural competence[2]. Despite the low levels of sequence homology, RRNPP members show a similar architecture consisting of a C-terminal peptide-binding domain and an N-terminal effector domain that, with the exception of the Rap subfamily, possess DNA-binding capacity. The peptide-binding domain consists of 5-9 tetratricopeptide repeats (TPR) that adopt a superhelical fold with a concave inner groove where the regulatory peptide is accommodated[2]. Binding of the specific peptide to the TPR domain induces conformational changes that regulate the activity of the effector domain but strikingly for proteins with similar architecture, the specific allosteric changes vary amongst receptors even within the same subfamily[3].

Recently, it has been found that phages can also exploit QS to communicate and make collective decisions. As a case in point, a novel system termed arbitrium, has been described in the *Bacillus subtilis* SPbeta group of phages. This system uses a six amino acid (aa) peptide as a communication signal and plays a key role in lysis–lysogeny decisions during infection[4]. The arbitrium system is composed of three genes; *aim*P, which encodes the arbitrium peptide, *aim*R, encoding a transcription factor that recognises and binds to AimP, and *aim*X, which exerts a negative regulatory effect on lysogeny, thus promoting lysis by a mechanism that is poorly understood[4,5]. AimR is a transcription factor that in its apo peptide-free form promotes *aim*X expression. In the initial stages of infection when the number of active phages is low, the arbitrium peptide is absent and AimR activates *aim*X expression, which maintains the lytic cycle of the phage. The arbitrium peptide is initially synthesised as a ~40 aa pro-peptide that via a secretion-internalisation process generates the mature AimP. During phage replication AimP will accumulate in the medium, increasing the intracellular concentration of mature AimP peptide until it reaches the threshold level required to bind to its cognate AimR receptor. Since AimR can no longer activate *aim*X transcription, a switch to the lysogenic cycle occurs thus preventing the killing of the entire bacterial population by the phage[4]. This simple, direct and elegant communication system allows phages to decide between lytic and lysogenic life cycles after infection. Importantly, it has been recently described that the arbitrium system is also essential for prophage induction[6,7].

It is assumed that AimR receptors from different SPbeta phages are specifically regulated by their cognate arbitrium peptide, suggesting that phages only communicate with their own progeny[4]. This proposition has a pivotal biological significance, because although homologous arbitrium systems were initially found in genomes of SPbeta-like phages, recent homology searches in the bacterial, archaeal and phages genomes have revealed that this type of communication system is widespread in the microbial world[5]. Arbitrium-like systems with the prototypical *aim*R-*aim*P organisation are found in many different types of phages, other mobile genetic elements (MGEs) and in the host bacteria. All of these systems can be clustered into nine clades according to their preferential peptide communication code plus a tenth clade that groups systems not present in MGEs, being considered as a possible non-arbitrium outgroup[5]. Although some clades seem to be exclusive to certain types of elements such as phages or MGEs, other clades encompass a variety of phages, MGEs and host bacteria[5]. Therefore, the arbitrium system could be a mechanism of communication among species.

Although the initial phylogenetic analyses did not identify AimR as a member of the RRNPP family[2], subsequent structural characterisation of the AimR from *B. subtilis* SPβ phage, after which the SPbeta family is named, confirmed that it is a bona fide member of this family[8]. We and others have recently obtained the crystal structures of the AimRs from the *B. subtilis* phages phi3T and SPβ (AimR$^{Phi}$ and AimR$^{SPβ}$, respectively), both in their apo and AimP-bound forms[8–12]. These studies have shown that these receptors have a HTH N-terminal DNA-binding domain (DBD) encompassing three helices followed by a C-terminal peptide-binding domain composed of nine highly degenerated TPR motifs. Moreover, our previous study identified the AimR-binding sites present in the *aim*X promoter region. Interestingly, the AimR-binding site exhibited a unique architecture composed of two six base-pair (bp) palindromic sequences separated by an unusually long 25 bp spacer[8]. The AimR$^{Phi}$ and AimR$^{SPβ}$ structures in complex with their specific AimPs showed an identical way of peptide recognition and binding. But noticeably, the mechanism of action proposed for AimP was somewhat different. For phi3T, the structural data obtained from an AimR$^{Phi}$ receptor with a double mutation showed that AimP induces AimR$^{Phi}$ dimer dissociation[9], thus preventing binding to the palindromic operator. However, recent in vitro data with the AimR$^{Phi}$ wild-type receptor showed that AimP does not undergo dimer-to-monomer conversion, with AimR$^{Phi}$ being a dynamic dimer with conformational states that are selectively stabilised by the target DNA or the inhibitory AimP peptide[13]. Similarly, the different structures solved for AimR$^{SPβ}$ confirm that the arbitrium receptor is constitutively a dimer and binding to its cognate peptide does not disrupt the dimeric organisation. Strikingly however, two mechanisms of action for AimP have been proposed for the AimR$^{SPβ}$ receptor. In the initial structures reported by other groups, the conformation exhibited by AimR$^{SPβ}$, both in its apo or AimP-bound forms, remained invariable[9–12]. By contrast, we have recently proposed that AimR$^{SPβ}$ undergoes drastic conformational changes upon peptide-binding[8]. In our studies, the overall structure of the individual AimR$^{SPβ}$ protomers are quite similar to those presented in previous studies, however huge differences in dimer arrangement have been demonstrated. These discrepancies have led us to suggest an alternative mechanism of AimR inhibition by the AimP peptide. The structures solved by our group support that apo AimR possesses a high degree of plasticity as a receptor. Once it recognises and binds to AimP, this interaction induces a compact and rigid conformation that approaches the AimR-DNA recognition helices, preventing them from reaching the distal operator palindromic sequences[8]. However, the alternative structures presented by other groups proposed a radically different mechanism based on the fact that AimR had an open conformation both in the apo and AimP-bound states. In the apo form, the authors proposed that binding to the DNA would force a more compact conformation allowing the AimR DBDs to bind to the palindromic sequences present in the *aim*X promoter region. Contrary to what we proposed, in this alternative mechanism the authors hypothesised that binding to AimP would stabilise the extended AimR conformation, a disposition that prevents the recognition helices to reach the distal boxes present in the *aim*X operator[10].

In this work, using multiple and complementary experimental approaches, we clarify this discrepancy and clearly establish the molecular mechanism by which AimP alters AimR function. More

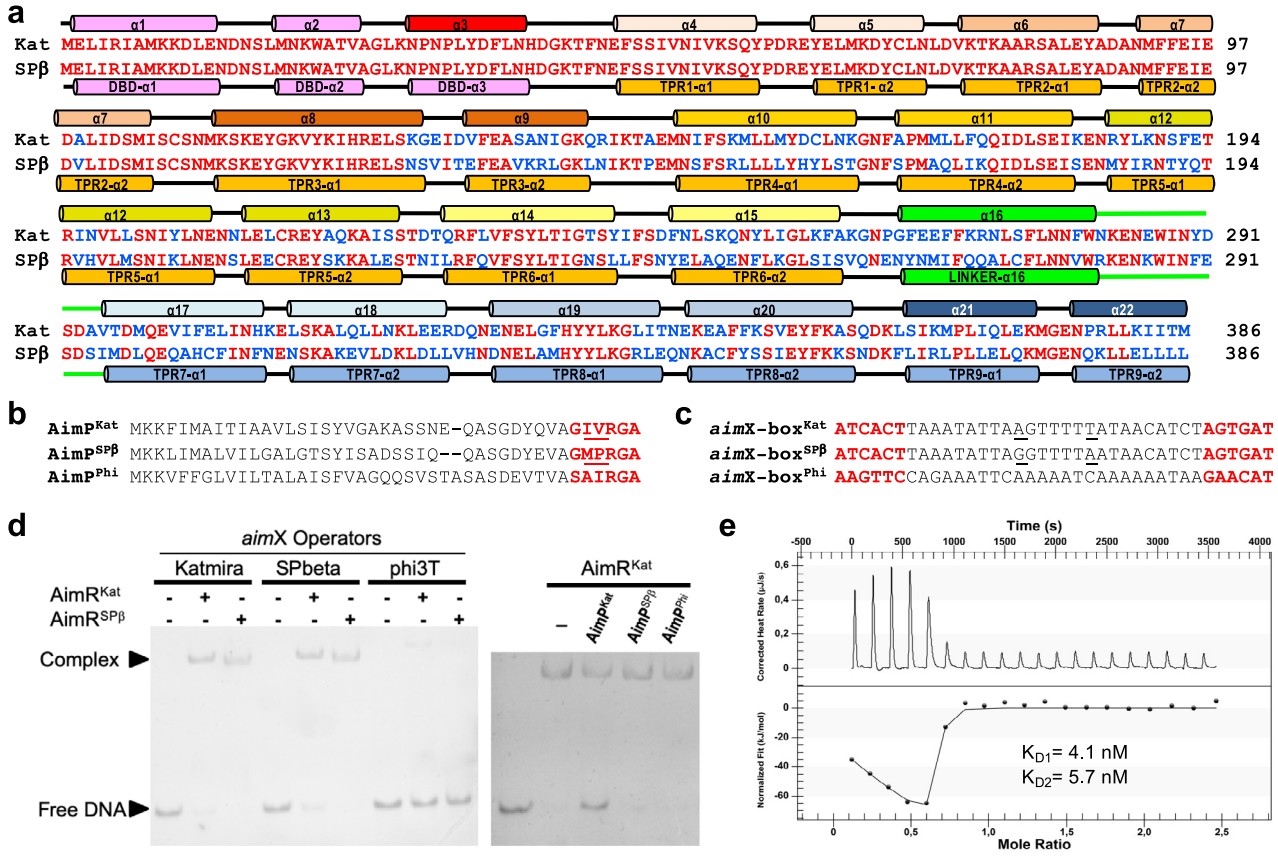

**Fig. 1 AimRs present chimeric traits. a** Sequence alignment of AimR[Kat] and AimR[SPβ]. Identical residues are in red colour. Structural elements are shown above and below of AimR[Kat] and AimR[SPβ] sequences, respectively, and labelled by helices for the former or domains for the latter. **b** Sequence alignment of AimP[Kat], AimR[SPβ] and AimP[Phi]. Mature AimPs are highlighted in red with differential positions between AimP[Kat] and AimR[SPβ] mature peptides underlined. **c** Sequences of DNA operators upstream of the putative *aim*X genes for AimR[Kat] (*aim*X-box[Kat]), AimR[SPβ] (*aim*X-box[SPβ]) and AimR[Phi] (*aim*X-box[Phi]), with the 6 bp inverted repeats highlighted in red. The two different positions between *aim*X-box[Kat] and *aim*X-box[SPβ] operator spacers are highlighted underlined. **d** EMSA analysis shown that both AimR[Kat] and AimR[SPβ] are able to recognise reciprocally their *aim*X operators but not the operator for phage phi3T. The AimR[Kat] binding to its operator is specifically disrupted by AimP[Kat]. EMSA assays have been repeated independently three times with similar results. Source data are provided as a Source Data file. **e** ITC measurement of AimR[Kat]-AimP[Kat] binding affinity. Thermogram is adjusted to two-binding site model and the two $K_D$ ($K_{D1}$ and $K_{D2}$) values, one per monomer in the AimR dimer, are shown.

importantly, our results reveal that AimR receptors have a unique domain organisation that can allow cross regulation among different AimRs. We anticipate these results may have crucial ecological and evolutionary implications in the microbial world.

## Results

**In silico characterisation of the arbitrium system present in the Katmira phage.** To answer the discrepancies in the proposed molecular mechanisms of action for the AimR of SPβ phage, we started this study by characterising a closely related arbitrium system. After an exhaustive database analysis searching for AimR[SPβ] homologues, we identified a very interesting chimeric AimR, present in a resident phage of *B. subtilis* KATMIRA1933 strain (hereafter Katmira phage), and recently also found in *Bacillus* phage vB_BsuS-Goe11[14]. AimR[SPβ] and the AimR from Katmira phage (AimR[Kat]) have a 66.9% sequence identity overall but this identity is not equally distributed. Instead, it is localised in specific regions of the proteins. In their N-terminal regions, including the DBD and the two initial TPRs (residues 1–125), AimR[Kat] and AimR[SPβ] presented almost identical sequences (only a single conservative change in TPR2). However, these proteins showed only 50% identity for the rest of the protein (residues 126–386) (Fig. 1a). This variable region corresponds to

the AimR[SPβ] TPRs 3–9 (Fig. 1a), which are involved in AimP recognition[8].

Since the C-terminal regions (TPRs 3–9) responsible for AimP recognition are not conserved in these AimR proteins, we hypothesised that the mature AimP peptides should be also different. The arbitrium system of Katmira phage belongs to clade 2 of clustered arbitrium systems identified during previous phylogenetic analyses[5]. In this clade, the mature form of AimP has been proposed to be 6 residues long[5]. Based on the characterisitics defined for AimP[5,8], as well as the predicted localisation of *aim*P in the phage genome, we proposed that the active Katmira AimP peptide (AimP[Kat]) would correspond to GIVRGA (Fig. 1b). This peptide differs in two amino acids from GMPRGA, the SPβ AimP peptide (AimP[SPβ]; Fig. 1b)[4,8].

In SPβ phages, *aim*X localises 3′ of *aim*P. We therefore scrutinised this region in the Katmira phage genome to identify the DNA-binding region recognised by AimR[Kat]. A putative AimR[Kat] binding box, consisting of 6 bp inverted repeats separated by 25 bp (ATCACT-25-AGTGAT), was identified (Fig. 1c) downstream of *aim*P[Kat] (*aim*X-box[Kat]). Interestingly, and in support of the fact that both AimR[SPβ] and AimR[Kat] have identical DBDs, the putative *aim*X-box[Kat] was almost identical to that found for AimR[SPβ] downstream of *aim*P[SPβ] (*aim*X-box[SPβ]) aside from two substitutions in the spacer region[8]. Further, both

**Table 1 Biolayer interferometry kinetic analysis of AimR receptors binding to *aim*X operators.**

| Receptor | DNA operator | $K_D$ (M)[a] | $k_{on}$ (M$^{-1}$s$^{-1}$) | $k_{off}$ (s$^{-1}$) | X2 | R2 |
|---|---|---|---|---|---|---|
| AimR$^{SP\beta}$-I[b] | *aim*X-box$^{SP\beta}$ | $1.93 \times 10^{-8}$ | $3.75 \times 10^{6} \pm 3.24 \times 10^{5}$ | $7.26 \times 10^{-2} \pm 2.29 \times 10^{-3}$ | 3.98 | 0.99 |
| AimR$^{SP\beta}$-I | *aim*X-box$^{Kat}$ | $1.96 \times 10^{-8}$ | $3.98 \times 10^{6} \pm 4.27 \times 10^{5}$ | $7.80 \times 10^{-2} \pm 2.50 \times 10^{-3}$ | 2.79 | 0.99 |
| AimR$^{SP\beta}$-I | *aim*X-box$^{Phi}$ | NBD[d] | | | | |
| AimR$^{Kat}$ | *aim*X-box$^{SP\beta}$ | $2.14 \times 10^{-8}$ | $3.43 \times 10^{6} \pm 3.05 \times 10^{5}$ | $7.32 \times 10^{-2} \pm 2.25 \times 10^{-3}$ | 4.78 | 0.99 |
| AimR$^{Kat}$ | *aim*X-box$^{Kat}$ | $2.31 \times 10^{-8}$ | $3.26 \times 10^{6} \pm 2.33 \times 10^{5}$ | $7.52 \times 10^{-2} \pm 1.90 \times 10^{-3}$ | 3.35 | 0.98 |
| AimR$^{Kat}$ | *aim*X-box$^{Phi}$ | NBD | | | | |
| AimR$^{SP\beta}$—II[c] | *aim*X-box$^{SP\beta}$ | $1.16 \times 10^{-7}$ | $6.31 \times 10^{5} \pm 2.88 \times 10^{4}$ | $7.35 \times 10^{-2} \pm 1.23 \times 10^{-3}$ | 4.19 | 0.99 |
| AimR$^{SP\beta}$—II | *aim*X-box$^{Kat}$ | $1.16 \times 10^{-7}$ | $6.77 \times 10^{5} \pm 1.82 \times 10^{4}$ | $7.71 \times 10^{-2} \pm 1.12 \times 10^{-3}$ | 1.76 | 0.99 |

[a]Values are the mean of five measurements.
[b]AimR$^{SP\beta}$-I: AimR$^{SP\beta}$ receptor without tags.
[c]AimR$^{SP\beta}$-II: AimR$^{SP\beta}$ receptor with C-terminal His tag.
[d]NBD: not binding detected in the experimental conditions used. $K_D > 1 \times 10^{-6}$ M.

*aim*X-box$^{Kat}$ and *aim*X-box$^{SP\beta}$ sites possess high adenine and thymine content, providing flexibility in this region (Fig. 1c).

The above results demonstrated that while both phages could bind to identical DNA operators, their AimRs are regulated by different peptides. Consequently, the observed variability in the TPR domains may have relevant biological implications, such as avoiding crosstalk among related phages.

**AimR$^{Kat}$ and AimR$^{SP\beta}$ recognise identical operators but are regulated by different peptides.** To test whether AimR$^{Kat}$ and AimR$^{SP\beta}$ bind to identical DNA boxes, carrying the structure ATCACT-25-AGTGAT, we performed electrophoretic mobility shift (EMSA) and biolayer interferometry (BLI) assays. The EMSA analyses using AimR$^{Kat}$ and AimR$^{SP\beta}$ and DNA fragments encompassing their cognate DNA-binding regions (*aim*X-box$^{Kat}$ and *aim*X-box$^{SP\beta}$) showed almost identical binding profiles (Fig. 1d). As expected, both AimRs recognised both DNA fragments with identical affinity. This was confirmed with the BLI analyses (affinities in the range 19–23 nM) (Table 1). However, neither recognised the AimR-binding box from phage phi3T (*aim*X-box$^{Phi}$) used as negative control (Fig. 1c, d, Table 1), confirming the specificity of the observed interactions between AimR$^{Kat}$ and AimR$^{SP\beta}$ receptors and their DNA targets.

To test whether GIVRGA is the regulatory AimP$^{Kat}$ peptide, we analysed its binding to AimR$^{Kat}$ using isothermal titration calorimetry (ITC). The experiment showed a biphasic thermogram, suggesting an allosteric effect between the two AimP-binding sites, one in each AimR$^{Kat}$ monomer, within the dimer. However, the peptide GIVRGA binds to the AimR$^{Kat}$ dimer with a similar high affinity at both peptide sites ($K_D$ values, $K_{D1}$ 4.1 ± 3.2 nM and $K_{D2}$ 5.7 ± 2.1 nM for monomer 1 and monomer 2, respectively), supporting a weak but existing cooperativity between the two AimP-binding sites on the dimeric AimR receptor (Fig. 1e). Finally, we analysed the ability of AimP$^{Kat}$ to disrupt the AimR$^{Kat}$-DNA interaction. Our EMSA assays clearly showed that AimP$^{Kat}$ induced the release of AimR$^{Kat}$ from its DNA operator, while neither GMPRGA (AimP$^{SP\beta}$) nor SAIRGA (the phi3T AimP peptide, AimP$^{Phi}$) (Fig. 1b) affected AimR$^{Kat}$ in its ability to recognise its cognate DNA box (Fig. 1d). Taken together, these results have identified and characterised the AimR and AimP components of the arbitrium system from Katmira phage, including the regulatory activity and specificity of AimP$^{Kat}$.

**In vivo characterisation of the AimR$^{Kat}$-AimP$^{Kat}$ pair.** It has been recently demonstrated that AimR is required for induction of the SPβ prophage[6,7]. We showed that after treatment of the wild-type (wt) and *aim*R-mutant prophages with mitomycin C (MC), which activates the bacterial SOS response, the

*aim*R-mutant showed a 50× reduction in the phage titre compared to that observed with the wt phage along with a cloudy-diffuse morphology of the plaques[6] (Fig. 2). This defect can be complemented when the *aim*R$^{SP\beta}$ is expressed from plasmid pDR110. Since our previous results indicated that both AimR$^{Kat}$ and AimR$^{SP\beta}$ recognise the same AimR box in the phage genomes, we hypothesised that expression of AimR$^{Kat}$ from pDR110 would also complement the SPβ Δ*aim*R prophage. Indeed, AimR$^{Kat}$ complemented the SPβ Δ*aim*R mutant and the behaviour of the mutant phage complemented with the non-cognate AimR$^{Kat}$ was indistinguishable to that observed when the mutant was complemented with its cognate AimR$^{SP\beta}$ (Fig. 2a). Moreover, when a lysate of SPβ Δ*aim*R was used to infect complemented recipient strains with either AimR$^{SP\beta}$ or AimR$^{Kat}$, the morphology of the plaques showed a "sharper" phenotype in both cases, further showing complementation of the mutant (Fig. 2b).

Next, and to confirm the role of the AimP$^{Kat}$ in vivo, we repeated the previous experiment in which the SPβ Δ*aim*R prophage was MC induced in the presence of AimR$^{Kat}$ or AimR$^{SP\beta}$, now with the addition of their cognate peptides, AimP$^{Kat}$ or AimP$^{SP\beta}$, respectively. The corresponding peptides were supplemented into the growth medium before MC-induction of the different prophages and after 12 h, the phage titres were quantified. In support of the previous results, addition of AimP$^{SP\beta}$ reduced the titre of the mutant phage complemented with AimR$^{SP\beta}$ (10 times)[6], confirming that the formation of the AimP-AimR complex interferes with prophage induction (Fig. 2a). An identical result was observed when AimP$^{Kat}$ was added to the mutant strain expressing AimR$^{Kat}$, therefore confirming the role of AimP$^{Kat}$ as the mature Katmira AimP (Fig. 2a).

**Crystal structure of apo AimR$^{Kat}$.** Once we identified and partially characterised the components of the arbitrium system present in the Katmira phage, and in order to shed light on the mechanism of action for these receptors, we tried to establish the molecular basis of the AimR$^{Kat}$-AimP$^{Kat}$ interaction. To do this, we solved the crystal structure of AimR$^{Kat}$ in its apo and peptide-bound forms. AimR$^{Kat}$ apo crystals diffract to 2.4 Å and belong to space group P2$_1$2$_1$2$_1$ showing two molecules in the asymmetric unit (ASU) (Supplementary Table 1). Both protomers are almost identical with an average RMSD (root mean square deviation of Cα carbon coordinates superimposition) of 0.7 Å. The protomer fold was similar to that observed for AimR$^{SP\beta}$, consisting of an N-terminal three helical (α1–α3) HTH DBD followed by a C-terminal regulatory domain composed of 19 helices (α4–α22) arranged as 9 degenerated TPRs (Figs. 1a, 3a). These TPR domains can be further divided into two subdomains, the TPR$^{N-ter}$ (TPR1-6) and the TPR$^{C-ter}$ (TPR7-9) connected by a

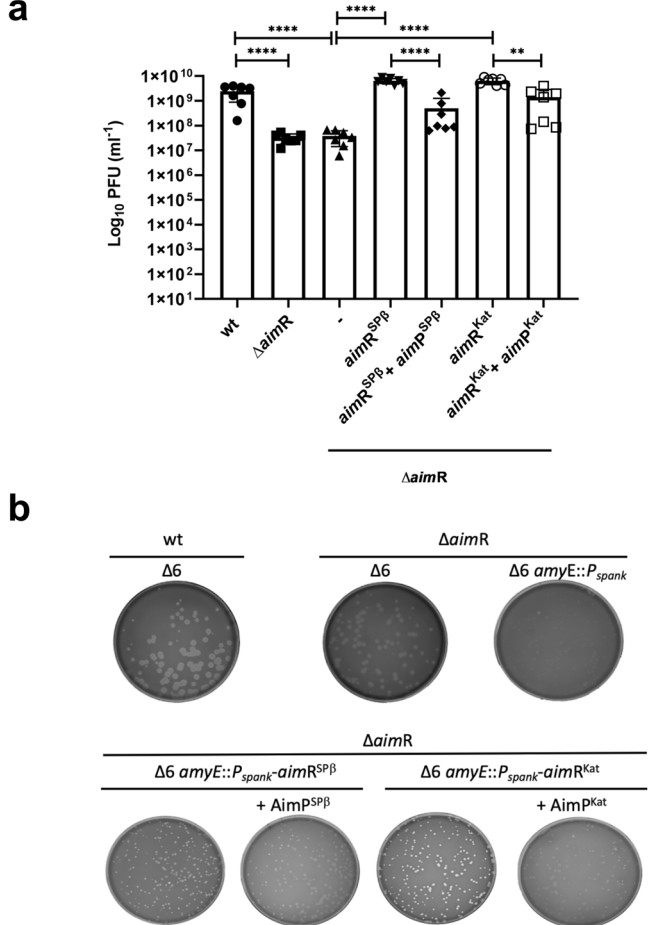

**Fig. 2 In vivo characterisation of SPβ and Katmira arbitrium systems.**
**a** Complementation of SPβ ΔaimR by aimR$^{SPβ}$ and aimR$^{Kat}$. Strains lysogenic for phage SPβ wt, ΔaimR and ΔaimR with the pDR110 cloned gene of aimR SPβ and Katmira were MC induced (0.5 μg/ml) and the number of resulting phages were quantified by titering using B. subtillis 168 Δ6 as the recipient strain. The complemented strains were induced with IPTG and when indicated 5 μM of peptide AimP$^{SPβ}$ or AimP$^{Kat}$ was added. The results are represented as the plaque forming units (PFUs) ml⁻¹. The means and SDs are presented for 7 independent repeats (n = 7). An ordinary one-way ANOVA of transformed data followed by a Tukey's multiple comparisons test was performed to compare mean differences between titres. Adjusted p values were as follows: ****$p ≤ 0.0001$; **$p = 0.017$. Source data are provided as a Source Data file. **b** Plaque morphology of SPβ wt and ΔaimR phages using different receptor strains. Lysates from phage SPβ wt and ΔaimR were used to titre into B. subtilis 168 Δ6 and Δ6 with the Pspank cloned gene of aimR SPβ and Katmira as the recipient strain. A dilution of these lysates was performed to visualise around 200 pfu. When indicated 5 μM of peptide AimP$^{SPβ}$ or AimP$^{Kat}$ was added before plating. The resulting plaque morphologies were photographed.

linker region of 30 residues that includes one α helix (α16) (Figs. 1a, 3a). The two protomers form a dimer in the crystal (Fig. 3b), an oligomeric organisation confirmed in solution by size-exclusion chromatography with multi-angle light scattering (SEC-MALS) (Supplementary Fig. 1a). The dimeric AimR$^{Kat}$ conformation was similar to that observed in the AimR$^{SPβ}$ apo structures previously described by our group[8] (hereafter AimR$^{SPβ}$-I), but differs in the dimerisation surface with the AimR$^{SPβ}$ structures reported by other groups[9–12] (Supplementary Figs. 2, 3). The three apo structures of the AimR$^{SPβ}$ reported by these groups (PDBs 6IPX, 6JG5 and 5ZW5) are virtually identical (Supplementary Fig. 2), in both the

individual protomers (RMSD ~0.4 Å for superimposition of 390 Cα residues) and the dimers (RMSD ~1.6 Å for 768 residues). Besides this, 6IPX and 6JG5 structures were generated from crystals showing the same space group (P2₁2₁2), almost identical unit cells and similar crystallisation conditions (Supplementary Fig. 2), indicating that they correspond to the same crystal form. Therefore, since it was solved to higher resolution, we selected the PDB 6JG5 as a representative of this AimR$^{SPβ}$ conformation (hereafter AimR$^{SPβ}$-II) for further structural comparisons. While the AimR$^{SPβ}$ protomers in the AimR$^{SPβ}$-II structures are quite similar to those observed in the apo AimR$^{SPβ}$-I and AimR$^{Kat}$ structures (RMSDs < 0.9 Å for protomers superimposition), the conformation of the AimR$^{SPβ}$-II dimer varies dramatically among these two groups (Supplementary Fig. 3). AimR$^{Kat}$, AimR$^{SPβ}$-I and AimR$^{Phi}$ structures show dimers with large contact interfaces spanning more than 1420 Å² in AimR$^{Kat}$. However, the dimerisation surface is reduced by a half (760 Å²) in the AimR$^{SPβ}$-II model, which is a consequence of the separation of the two protomers (Supplementary Figs. 2, 3). We have therefore named the AimR$^{SPβ}$-II conformation as the "open" dimer by contraposition of the "closed" dimer conformation observed in AimR$^{Kat}$, AimR$^{SPβ}$-I and AimR$^{Phi}$ apo structures. The largest interface in the closed dimer structure can be divided into two interacting areas, one located at TPR$^{C-ter}$ subdomain involving residues from TPR8 (α20) and TPR9 C-terminal capping helix (α22) and the other one located at the TPR$^{N-ter}$ subdomain generated by residues from TPR3 (α9 and α10) and TPR4 (α11) (Fig. 3b, Supplementary Fig. 4). While both structures show highly similar interactions in the TPR$^{C-ter}$ area, highlighting a pseudo-leucine zipper generated by the C-terminal capping helices, the interactions in the TPR$^{N-ter}$ area vary between the structures (Supplementary Figs. 2, 3). This observation confirms our initial hypothesis that the TPR$^{N-ter}$ area acts as a "slipping" contact interface that allows the dimer plasticity observed in the structures of apo AimR$^{SPβ}$-I[8]. Oppositely, dimerisation in the AimR$^{SPβ}$-II open dimer conformation is only mediated by the C-terminal conserved surface and the slipping surface is solvent exposed and does not participate in dimer formation (Supplementary Fig. 2). This was a striking discrepancy for structures of the same protein, such as the apo AimR$^{Phi}$ structure (PDB 5ZVV)[9], a distantly related AimR receptor which has 40% identity with AimR$^{SPβ}$, showing the closed dimer and presenting both the conserved C-terminal leucine zipper and the variable slipping dimerisation surfaces (Supplementary Fig. 2).

**AimP induces a compact conformation of AimR.** To analyse the molecular mechanism of AimR$^{Kat}$ peptide inhibition we attempted to co-crystallise AimR$^{Kat}$ with AimP$^{Kat}$. The structure of the complex (AimR-AimP$^{Kat}$) was determined at a resolution of 2.7 Å by molecular replacement (Supplementary Table 1) using the structure of AimR-AimP$^{SPβ}$ complex (PDB 6PH5) as a model. The AimR-AimP$^{Kat}$ asymmetric unit contains two AimR molecules arranged as a dimer, with one AimP$^{Kat}$ bound to the TPR domain of each AimR$^{Kat}$ protomer (Fig. 4a). The dimeric organisation in solution for the AimR-AimP$^{Kat}$ complex was confirmed by SEC-MALS (Supplementary Fig. 1). Having the structures of AimR$^{Kat}$ in its apo and AimP$^{Kat}$-bound conformations, we not only analysed the mechanism of peptide action but also whether it corresponds to any of the two alternative mechanisms proposed for AimR$^{SPβ}$. This comparison showed that the peptide induces a closure movement of AimR protomer, produced by a ~15° relative rotation in the TPR$^{N-ter}$ subdomain (calculated with Dyndom[15]). This results in a more compact structure with a shorter distance between the TPR$^{N-ter}$ and TPR$^{C-ter}$ subdomains, reducing the TPR superhelix pitch up to 5 Å (Fig. 4b). Further, due to the reduced proximity of the subdomains, new interactions are produced between TPR$^{N-ter}$ and

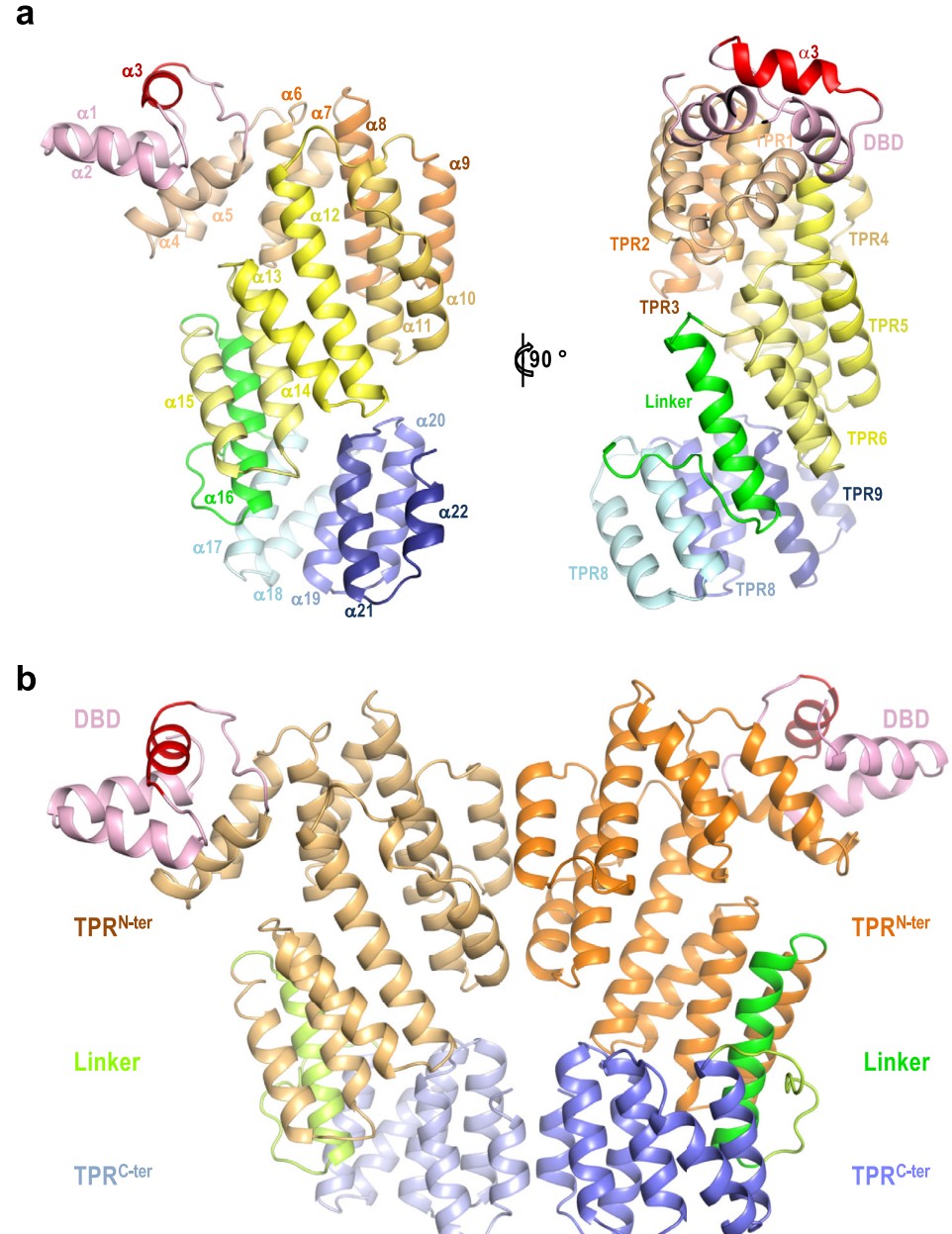

**Fig. 3 Architecture of apo AimR^Kat. a** Two orthogonal views of apo AimR^Kat protomer in cartoon rendering. Structural elements are coloured as in Fig. 1a and labelled by helix and TPRs in right and left representations, respectively. **b** Cartoon rendering of biological dimeric AimR^Kat in apo state. Functional domains are labelled and coloured pink (DBD), orange (TPR^N-ter) green (linker) and blue (TPR^C-ter) with darker tones in the right protomer. The DNA recognition α3 helices are coloured in red and in (**a**).

TPR^C-ter that tie the protomer in the compact conformation. Thermal shift assays of AimR^Kat revealed a huge 20 °C increment to the melting temperature (Tm) in the presence of AimP^Kat, confirming that the cognate peptide strongly stabilises AimR^Kat. Conversely, AimP^SPβ or AimP^Phi, used as control peptides, did not have any effect (Supplementary Fig. 5a). Similarly, we have previously reported that AimP^SPβ strongly increases (10 °C) the Tm of AimR^SPβ, supporting the idea that AimP induces a similar conformational compaction that stabilises this type of AimR receptor[8]. Indeed, the reporter comparison for AimR^SPβ-I in its apo and AimP^SPβ-bound structures (PDB 6PH5; hereafter AimR-AimP^SPβ-I) showed a similar closing movement for the protomers in the presence of the peptide (reduction of ~5 Å in the superhelix pitch) resulting in more structurally compact AimR

protomers[8] (Supplementary Fig. 6). However, and as occurred with the apo forms, three alternative structures of AimR^SPβ in complex with AimP^SPβ (PDBs 6IM4, 6JG9 and 5ZW6) have been reported by the same groups that solved the AimR^SPβ-II structures[9–12] (Supplementary Fig. 2). These three structures are almost identical to each other (RMSDs ~0.4 and ~1.9 Å for protomers and dimers superimposition, respectively). Indeed, two of these structures (PDBs 6JG9 and 6IM4) were obtained from crystals with the same space group and almost identical cell dimensions, and the third one (PDB 5ZW6) was crystallised in the same space group with similar cell dimensions (Supplementary Fig. 2), indicating that all of them correspond to the same structure. Therefore, we name this AimR^SPβ peptide-bound conformation as AimR-AimP^SPβ-II, selecting PDB 6JG9 as the

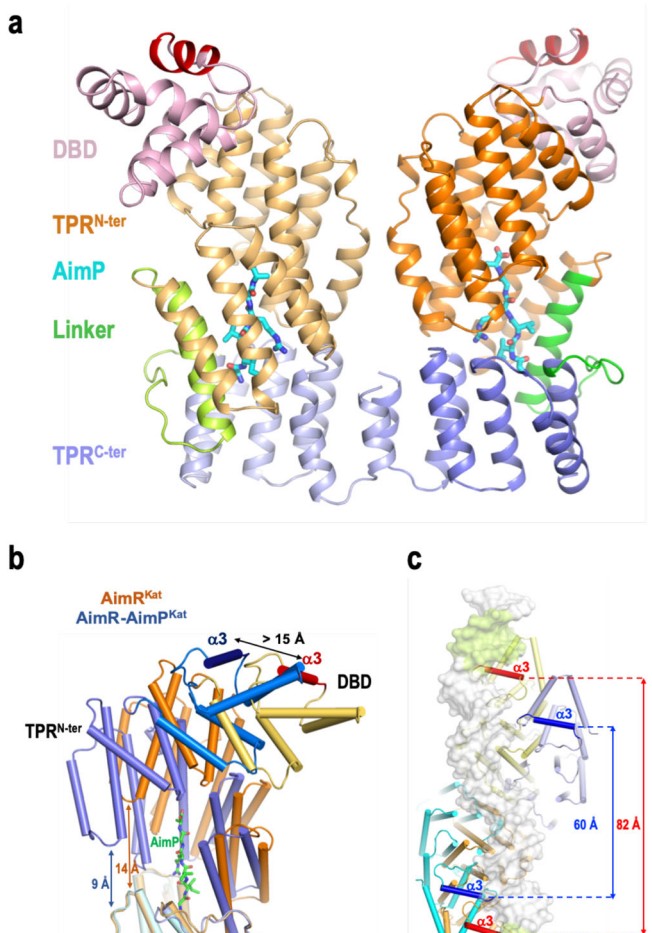

**Fig. 4 AimP-induced conformation changes in AimR. a** Cartoon rendering of dimeric AimP-AimR$^{Kat}$ complex. The structure is presented in similar view and colours as in Fig. 2b. AimPs are shown in sticks representation with carbon, oxygen and nitrogen atoms coloured in cyan, red and blue, respectively. **b** AimP induces a compact conformation of AimR. Superimposition of AimR$^{Kat}$ protomers in its apo (orange colours) and AimP-bound (blue colours) states shows how the TPR$^{N-ter}$ (dark hue) and TPR$^{c-ter}$ (light hue) domains approaches as the peptide binds reducing their distance from 14 to 9 Å. This compaction movement promotes a displacement greater than 15 Å in the recognition helix α3. AimR are rendered with helices as cylinders and the AimP in sticks representation with carbon coloured in green. **c** AimP binding prevents DNA operator recognition. Superposition of AimR$^{Kat}$ in its apo and AimP-bound conformations on the structure of AimR$^{SPβ}$ bound to its DNA operator (PDB 6HP7). A view from the DNA side with showing AimR$^{Kat}$ structures in cartoon rendering as in (**b**) and the operator DNA on white semi-transparent surface with the palindromic sequences highlighted in green. The recognition helices are highlighted in a darker colour and intra-dimeric distance is indicated.

representative of the group for further analysis because it was solved to higher resolution. Contrary to what was observed for our AimR$^{Kat}$ and AimR$^{SPβ}$-I structures, which suffered an important change in the presence of their cognate AimP peptides, both AimR$^{SPβ}$-II and AimR-AimP$^{SPβ}$-II showed identical "open" conformations (RMSD of 0.4 and 1.4 Å for the superposition of protomer and full dimer, respectively) (Supplementary Figs. 2, 6), suggesting that peptide binding would have null structural impact

on AimR$^{SPβ}$ structure. This observation, although surprising, could be anticipated since both AimR$^{SPβ}$-II and AimR-AimP$^{SPβ}$-II structures were obtained from almost identical crystals (Supplementary Fig. 2), indicating that the same conformation had been crystallised.

Importantly, the fact that both the apo and the AimP-bound forms present identical conformations makes it almost impossible to deduce the mechanism of peptide inhibition from the analysis of the AimR$^{SPβ}$-II structures. By contrast, the comparison of our AimR$^{Kat}$ and AimR$^{SPβ}$-I structures in their apo and peptide-bound states show similar conformational changes (Fig. 4b, Supplementary Fig. 6) that explain their conserved mechanism of action. As a result of the protomer compaction induced by the peptide, the dimer reorganises by altering the relative disposition of the DBDs, which then displace and approach each other. The distance between the DNA recognition helices (α3) in the AimR-AimP$^{Kat}$ dimer is only 60 Å, a reduction of more than 20 Å in comparison to the apo AimR$^{Kat}$ form (Fig. 4c). This separation between the two DBDs and their relative disposition is too small to allow them to interact with the two palindromic sequences present in the AimR operator which are separated ~85 Å due to the 25 bp spacer, as confirmed by the superimposition of AimR$^{Kat}$ and AimR-AimP$^{Kat}$ in the structure of AimR$^{SPβ}$ bound to its DNA operator (Fig. 4c). Further, this structural comparison also confirms that AimR$^{Kat}$ in its apo state exists in a DNA-binding competent conformation, similar to the DNA-bound AimR$^{SPβ}$ structure (RMSD of only 0.87 Å for the comparison of both dimers) (Supplementary Fig. 7). A similar mechanism of peptide action was observed for AimR$^{SPβ}$-I, where the binding of AimP to its cognate AimR reduced the distance of the two DBDs by ~10 Å[8]. The comparison of the AimR$^{Kat}$ and AimR-AimP$^{Kat}$ structures also confirmed that the peptide-induced dimer reorganisation is facilitated by the double dimerisation area observed in AimR$^{Kat}$ and AimR$^{SPβ}$-I. While the contacts between the TPR$^{C-ter}$ subdomain remain constant in the apo and peptide-bound conformations, governed by the pseudo-leucine zipper of the capping helix, we observe a TPR$^{N-ter}$ dimerisation surface change, irrespective of the fact that the residues mediating the interactions in this area are almost the same (Supplementary Fig. 4). This supports the role of the "slipping surface" provided by the TPR$^{N-ter}$ to allow the plasticity that AimR receptors require for their mechanism of action. Contrastingly, only the contacts mediated by the TPR$^{C-ter}$ subdomain are observed in the almost identical AimR$^{SPβ}$-II and AimR-AimP$^{SPβ}$-II structures, where the slipping surface is solvent exposed (Supplementary Figs. 2, 3, 6).

**Crystal structure of AimR$^{Kat}$-DNA complex confirms the mechanism of action.** In order to validate our proposed mechanism of action, we solved the crystal structure of AimR$^{Kat}$ in complex with a 45 bp DNA fragment (AimR$^{Kat}$-DNA) which includes the *aim*X-box$^{Kat}$ with the two 6 bp inverted repeats recognised by DBD domain (identical to the ones recognised by AimR$^{SPβ}$) and a 25 bp spacer. AimR$^{Kat}$-DNA crystals belong to space group P2$_1$ and diffracted to a resolution of 2.5 Å (Supplementary Table 1). The crystal structure was solved by molecular replacement using the structure of AimR$^{SPβ}$ in complex with DNA (AimR$^{SPβ}$-DNA; PDB 6HP7) as a model. The AimR$^{Kat}$-DNA asymmetric unit contains two AimR molecules arranged as a dimer and one duplex *aim*X-box$^{Kat}$ molecule (Fig. 5a). Overall, the structure is similar to AimR$^{SPβ}$-DNA (RMSD ~2.1 Å), how-ever, RMSD differences are not constant throughout the entire molecule. RMSD differences are minimal in the N-terminal part, which is bound to the DNA, and increase towards the C-terminal domain (Fig. 5b). As could be anticipated from the sequence

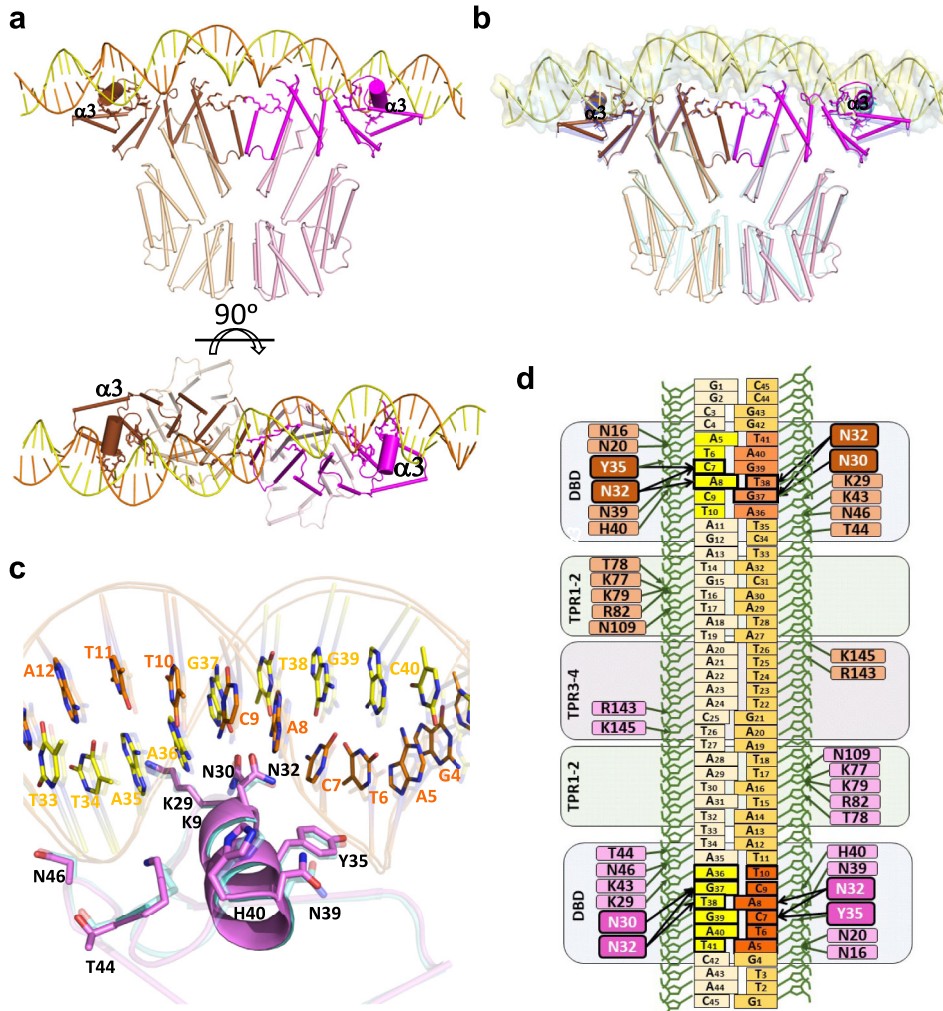

**Fig. 5 AimR^Kat DNA-binding characterisation. a** Overall structure of AimR^Kat in complex with DNA. Two orthogonal views are shown. Subunits A and B are represented in cartoon with helices as cylinders and coloured in brown and magenta respectively. DNA is painted in orange and yellow for 5′-3′and 3′-5′ strands, respectively. Darker colours have been used for DNA interacting regions with helix α3 shown with a bigger radius and DNA interacting residues represented in stick. **b** Overall superposition of AimR^Kat (same colour code as for **a**) and AimR^SPβ (coloured in blue). DNA is shown as yellow cartoon and surface for AimR^Kat and blue surface for AimR^SPβ. **c** Detail view of DNA specific recognition mediated by helix α3 in AimR^Kat (coloured in magenta) and AimR^SPβ (semi-transparent and coloured in blue). Interacting residues are shown in sticks and labelled. **d** Schematic representation of the DNA-AimR^Kat contacts. The colour code is the same as for (**a**). Sequence-specific interactions for DNA readout are highlighted with thicker lines and the residues carrying out these interactions with more intense colours.

identity between AimR^Kat and AimR^SPβ (almost 100% for DBD, TPR1 and TPR2) and the identical inverted repeats in their target DNA boxes (Fig. 1a, c), both proteins bind to DNA using equivalent interactions (Fig. 5c). The α3 helix from DBD inserts into DNA major groove allowing specific DNA readout by side chains from residues N30, N32 and Y35 that interact with the $T_6C_7A_8/A_{36}G_{37}T_{38}$ portion from the ATCACT inverted repeat of *aim*X-box^Kat (Fig. 5c, d). In addition, DNA binding is stabilised by an extensive indirect readout of the DNA backbone at both the inverted repeats and the spacer region. The DBD is involved in the indirect readout of the inverted repeats by polar interactions of residues N16, N20, K29, Y35, N39, H40, K43, T44 and N46 with DNA backbone. Meanwhile residues located in TPR^N-ter are involved in the 25 bp spacer binding by mainly polar interactions. Specifically, residues K77, T78, K79, R82 (located in α6 from TPR-1), N109 (loop connecting α7 and α8), R143 and K145 (loop connecting α9 and α10) mediate hydrogen-bound salt bridges with the DNA phosphates along the entire length of the spacer

(Fig. 5d). Of all these residues, only R143 differs from the equivalent residue in AimR^SPβ, an Asn residue that also interacts with the DNA backbone. The high A-T content in the 25 bp spacer allows a small DNA bending of around 20° towards AimR interacting surface. AimR^Kat-DNA and AimR^SPβ-DNA structures visualise the capacity of recognition of similar operators despite being regulated by different peptides, confirming structurally the cross regulation mediated by these operators.

**The C-terminal His-tag disturbs the dimer interface in AimR.** After the complete structural characterisation of AimR from phages SPβ and Katmira, we tried to understand the origin of the discrepancies observed in the two models that currently explain AimR fuction. We realised that all the described AimR structures were obtained from proteins produced recombinantly, fused to N- or C-terminal tags. However, the N-terminal His-tags from AimR^SPβ-I and AimR^Kat and C-terminal GST-tag from AimR^Phi

were removed prior crystallisation. In contrast, the C-terminal His-tags from the AimR$^{SP\beta}$-II structures were not removed from the recombinant proteins before crystallisation. Therefore all the AimR$^{SP\beta}$-II structures included the C-terminal LEHHHHH or LEYAHHHHH (extra residues 387-390/392) extension that is partially visible in several of the structures with the 'open' conformation (Supplementary Fig. 2). In the AimR$^{SP\beta}$-II structures, the presence of an extra Leu (L387) from the tag enlarges the leucine zipper of the capping C-terminal helix that orchestrates AimR dimerisation. The tag E388 establishes intermolecular contacts with residues from the second protomer and some His present in the tag (up to three are visible in some AimR$^{SP\beta}$-II PDBs) are projected towards the slipping dimerisation surface, interacting with residues from TPR$^{N-ter}$ and TPR$^{C-ter}$ subdomains (Fig. 6a). The C-terminal tag acts as a pair of forceps disturbing the slipping dimer interface and inducing the dimer 'opening' observed in AimR$^{SP\beta}$-II structures (Fig. 6a and Supplementary Fig. 8). Strikingly, the monomers from AimR$^{SP\beta}$-II and AimR-AimP$^{SP\beta}$-II structures show almost identical conformation as the AimR-AimP$^{SP\beta}$-I (RMSDs 0.5–0.7 Å) and AimR-AimP$^{Kat}$ (RMSDs 0.8–1 Å), with the exception of the C-terminal capping helices that are displaced around 4.5 Å (Fig. 6a and Supplementary Fig. 9). Therefore, both the AimR$^{SP\beta}$-II and AimR-AimP$^{SP\beta}$-II structures correspond to the peptide-inhibited conformation of the AimR receptor, but the presence of the extra C-terminal tag following the capping helix disturbs the dimerisation and precludes the dimer to adopt the closed conformation observed in AimR-AimP$^{SP\beta}$-I and AimR-AimP$^{Kat}$. Indeed, it seems that the tag induces the dimer open conformation since it is also adopted in the absence of peptide (Supplementary Figs. 2, 6, 8). If our proposal is correct, the destabilisation of the dimerisation surface induced by the C-terminal tag should have an impact on the stability and activity of the AimR dimer. Indeed this is the case, and the thermal shift assays for AimR$^{SP\beta}$ carrying the C-terminal His-tag present in AimR$^{SP\beta}$-II showed a decrease of more than 9 degrees in its Tm versus the protein without the tag (53.6 °C for AimR$^{SP\beta}$-I vs. 44.7 °C for AimR$^{SP\beta}$-II) (Supplementary Fig. 5b), supporting the loss of dimerisation surface area that makes the dimer less compact and stable. To evaluate the effect C-terminal tag in AimR activity we measured by BLI the AimR$^{SP\beta}$-II DNA-binding affinity, obtaining a value of $K_D$ one order of magnitude lower than the $K_D$ calculated for AimR$^{SP\beta}$-I (Table 1). This result is in close agreement with the structural data available. Two structures have been reported for the AimR$^{SP\beta}$-DNA complex with similar close conformation (RMSD 2.1 Å for the dimers superimposition) showing both constant and slipping dimerisation surfaces[8,10], which are similar to the AimR$^{Kat}$-DNA reported here (Figs. 5b, 6b and Supplementary Fig. 8). Of these two structures of AimR$^{SP\beta}$ in complex with DNA, one was generated from the protein without a tag[8](PDB 6HP7) and the second one from the protein including the C-terminal poly-His[10](PDB 6JG8). Inspection of the tagged structures showed that only the tag for one monomer was visible and it protrudes outside the AimR dimer (Fig. 6b and Supplementary Fig. 8) and packs with the DNA of another complex in the crystal. Therefore, the DNA is required to expel the C-terminal tag from the AimR dimerisation interface in order to acquire the biological competent closed conformation for interaction through the slipping zone. This fact is reflected in the DNA-binding analysis since the comparison of the $k_{on}$ and $k_{off}$ constants for both AimR$^{SP\beta}$ proteins shows that while the $k_{off}$ is identical, the $k_{on}$ is five times lower in the case of AimR$^{SP\beta}$-II (Table 1), confirming that the presence of the C-terminal tag hampers AimR from acquiring the competent conformation before DNA binding. However, once acquired, for which this tag must be expelled from the dimerisation interface as shown in the AimR structure (Fig. 6 and Supplementary Fig. 8), the binding is not affected. In addition, this DNA-induced conformational change explains why the heterologous expression of the receptor with the C-terminal tail can rescue in vivo a deletion mutant[10,11] although its affinity for the target DNA is significantly lower.

**AimR-P structures provide clues in arbitrium peptide specificity.** Since AimR$^{SP\beta}$ and AimR$^{Kat}$ have identical DBDs but are controlled by different peptides, we hypothesised that the structural comparison of AimR$^{Kat}$ and AimR$^{SP\beta}$ in complex with their cognate peptides would shed light on how different AimRs discriminate between related peptides. As suggested by the structural superimposition of AimR-AimP$^{SP\beta}$ and AimR-AimP$^{Kat}$ protomer (RMSD 1.2 Å), both use an identical peptide recognition mechanism (Fig. 7a, Supplementary Fig. 9). In the AimR-AimP$^{Kat}$ structure, the peptide binds to the concave side of the channel formed by the TPR-like repeats, and it establishes hydrophilic interactions at both N- and C-terminal ends, anchoring AimP$^{Kat}$ in an extended conformation (Fig. 7a). The AimP$^{Kat}$ N-terminal is bound to AimR$^{Kat}$ Q299 and E300 side chains (TPR7) by a hydrogen bond and a salt bridge, respectively. On the opposite end, the C-terminal carboxylate establishes a salt bridge with the R228 side chain (TPR6). These residues, involved in anchoring the ends of the peptide, are highly conserved among the clade 2 family of arbitrium receptors[8], supporting an identical peptide accommodation in the TPR domain. A majority of TPRs possess side chains that provide a mainly hydrophobic cleft where AimP$^{Kat}$ is accommodated while the peptide main chain is stabilised by hydrogen bonds with AimR$^{Kat}$ N202 and N273 side chains (Fig. 7a).

AimP peptides present a highly conserved RGA C-terminal half[4,8]. The structures shown here confirm that the side chains of the RGA peptide region are recognised by identical residues in both AimRs (Fig. 7a). The AimP R4 side-chain interacts with AimR N206, N329 and D360, while the AimP A6 with V198 and L199 (Fig. 7a). Opposingly, the N-terminal halves of the peptides show differences at positions 2 and 3 (I2-V3 in AimP$^{Kat}$ vs. M2-P3 in AimP$^{SP\beta}$) and, consequently, the residues surrounding these AimP side chains also show differences between AimR$^{Kat}$ and AimR$^{SP\beta}$ (Fig. 7a). Thus, we propose these differences should confer specificity by enhancing or disturbing peptide binding. AimR$^{Kat}$ has an Asn at position 273 (N273) that anchors AimP$^{Kat}$ V3 by interacting with its N and O backbone. Conversely, in AimP$^{SP\beta}$ the Pro residue at this peptide position eliminates the hydrogen bridge within the peptide N backbone and also its rigid pyrrolidine side-chain would introduce steric hinderance with the AimR$^{Kat}$ N273 side chain (Fig. 7a). These differences seem to represent the most important contribution to the specificity of the peptide between both AimRs, since the second varible AimP side chain at position 2 (Ile vs. Met) is accommodated by both AimRs in a similar large hydrophobic pocket without major steric constrictions (Fig. 7a). To test this posibility, we generated the AimR$^{Kat}$ mutant form of N273 to Ala (AimR$^{Kat}$-N273A) to emulate AimR$^{SP\beta}$, and its AimP-binding capacity was tested. ThermoFluor assays showed that AimP$^{SP\beta}$ stabilises AimR$^{Kat}$-N273A but not AimR$^{Kat}$ (Fig. 7b), supporting the acquisition of binding capacity for this peptide. AimR$^{Kat}$-N273A still conserves the binding capacity to AimP$^{Kat}$ although slightly reduced as is suggested by the decrease in the Tm (Fig. 7b). Next, we quantified the changes in peptide affinity of AimR$^{Kat}$-N273A using ITC. As was observed for AimR$^{Kat}$, ITC titration of AimR$^{Kat}$-N273A showed biphasic thermograms that were fitted to a two-binding site model, one per monomer in the AimR dimer (Figs. 1e, 7c). In agreement with the ThermoFlour assays, AimR$^{Kat}$-N273A affinity for AimP$^{Kat}$ has been lowered 100 times ($K_D$ values 194 ± 4 and 356 ± 225 nM for monomer 1 and monomer 2, respectively) with

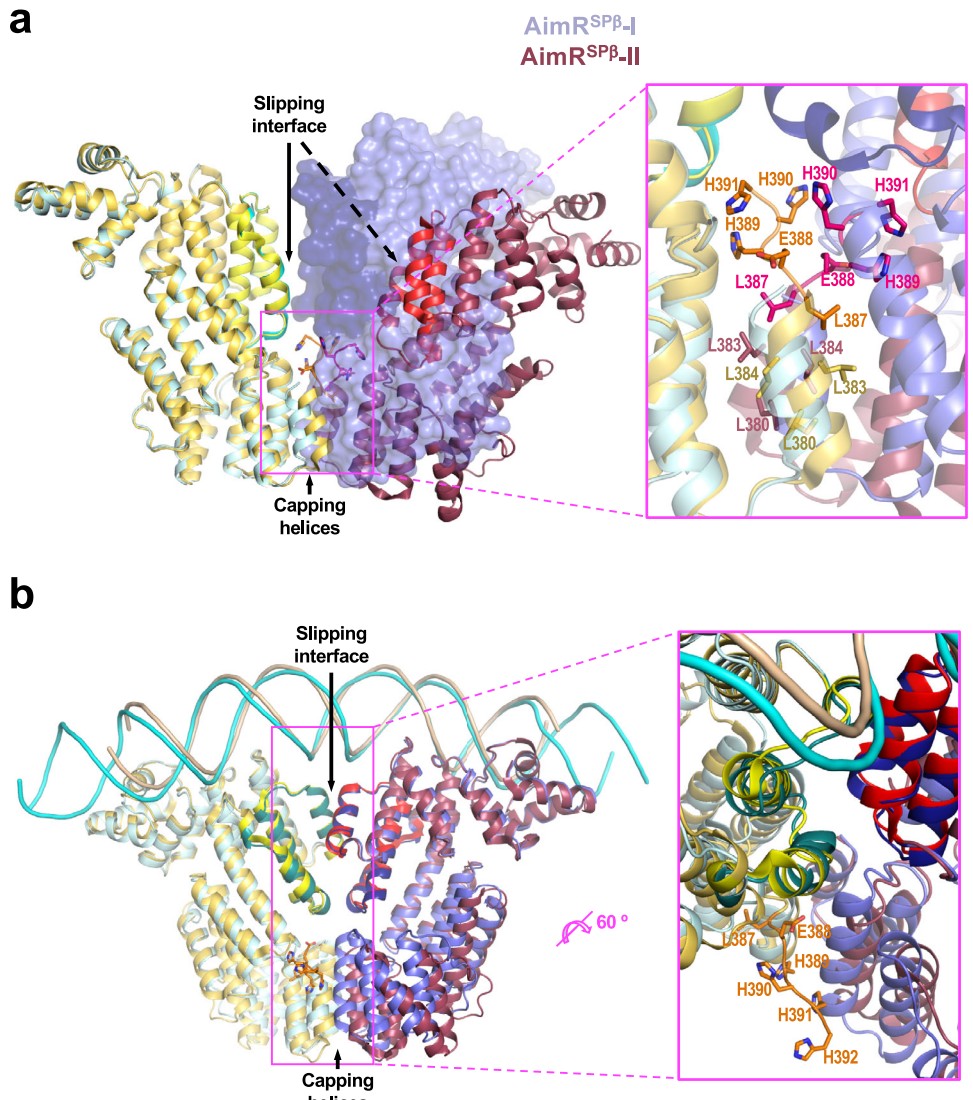

**Fig. 6 AimR dimerisation interference by C-terminal His tag.** Superimposed AimR$^{SP\beta}$-I (cyan-blue) and AimR$^{SP\beta}$-II (yellow-dark red) in its apo (**a**) and DNA-bound conformations (**b**). **a** The superposition of left protomers in the dimers in the apo state shows how the second protomer occupies different positions in each structure. While AimR$^{SP\beta}$-I maintains a closed conformation with the second protomer (on semi-transparent surface) interacting by two dimerisation interfaces (slipping and capping helices), in AimR$^{SP\beta}$ -II the second protomer (on cartoon) moves away presenting an open conformation where the slipping surfaces (red and light yellow) does not contact. A close view (right) shows how residues of the C-terminal His tag (residues 388-390) in AimR$^{SP\beta}$-II, shown on sticks with carbon atoms in the same colour as the corresponding protomer, are inserted in the dimerisation interface hampering the approach of protomers for interaction of slipping zones although capping helices interactions are maintained (side chains for hydrophobic interactions are shown in sticks and labelled). **b** The superimposition in the DNA-bound states shows identical DNA disposition (backbone representation) and AimR conformation (cartoon representation), using both dimerisation interfaces. A close view shows that the C-terminal His tag presents in AimR$^{SP\beta}$ -II only is visible for one protomer (side chains are shown in stick and labelled) and is projected into the solvent away from the dimerisation interface.

respect to wt AimR$^{Kat}$ and has acquired the capacity to bind AimP$^{SP\beta}$, although with low affinity (K$_D$ values $1.3 \pm 3.6$ and $1.6 \pm 6.6 \,\mu$M for monomer 1 and monomer 2, respectively) (Fig. 7c). However, AimR$^{Kat}$-N273A is unable to bind AimP$^{Phi}$ (Fig. 7c), supporting that the mutation does not confer non-specific binding capacity to regulatory peptides. Finally, we evaluated if the binding of AimR$^{Kat}$-N273A to the AimPs is functional. EMSA analyses showed that both AimP$^{SP\beta}$ and AimP$^{Kat}$, but not AimP$^{Phi}$, are able to induce the release of AimR$^{Kat}$-N273A from its target DNA in a similar way as AimP$^{Kat}$ induces release of AimR$^{Kat}$ (Fig. 7d).

**AimR receptor chimerity anticipates cross regulation.** The previous results confirm that the AimR receptors are modular

with an N-terminal DBD and a C-terminal regulatory domain. Peptide binding to the C-terminal domain induces a compact and rigid conformation that approaches the DBDs, hampering operator recognition. This separation of function in domains opens up ecological possibilities of great interest such as cross regulation between phages. As we have shown, AimR$^{Kat}$ and AimR$^{SP\beta}$ are regulated by different peptides but recognise the same target DNA, suggesting a possible cross regulation between them. So, we analysed if the cross regulation is shared by other members of SPbeta phage family. To test the crossregulatory hypothesis, we searched for AimR receptors that present high homology in the N-terminal DBD but recognise different AimP peptides, using SPβ and phi3T phages as "bait". Different homologues were identified for both AimRs in phages from *B.*

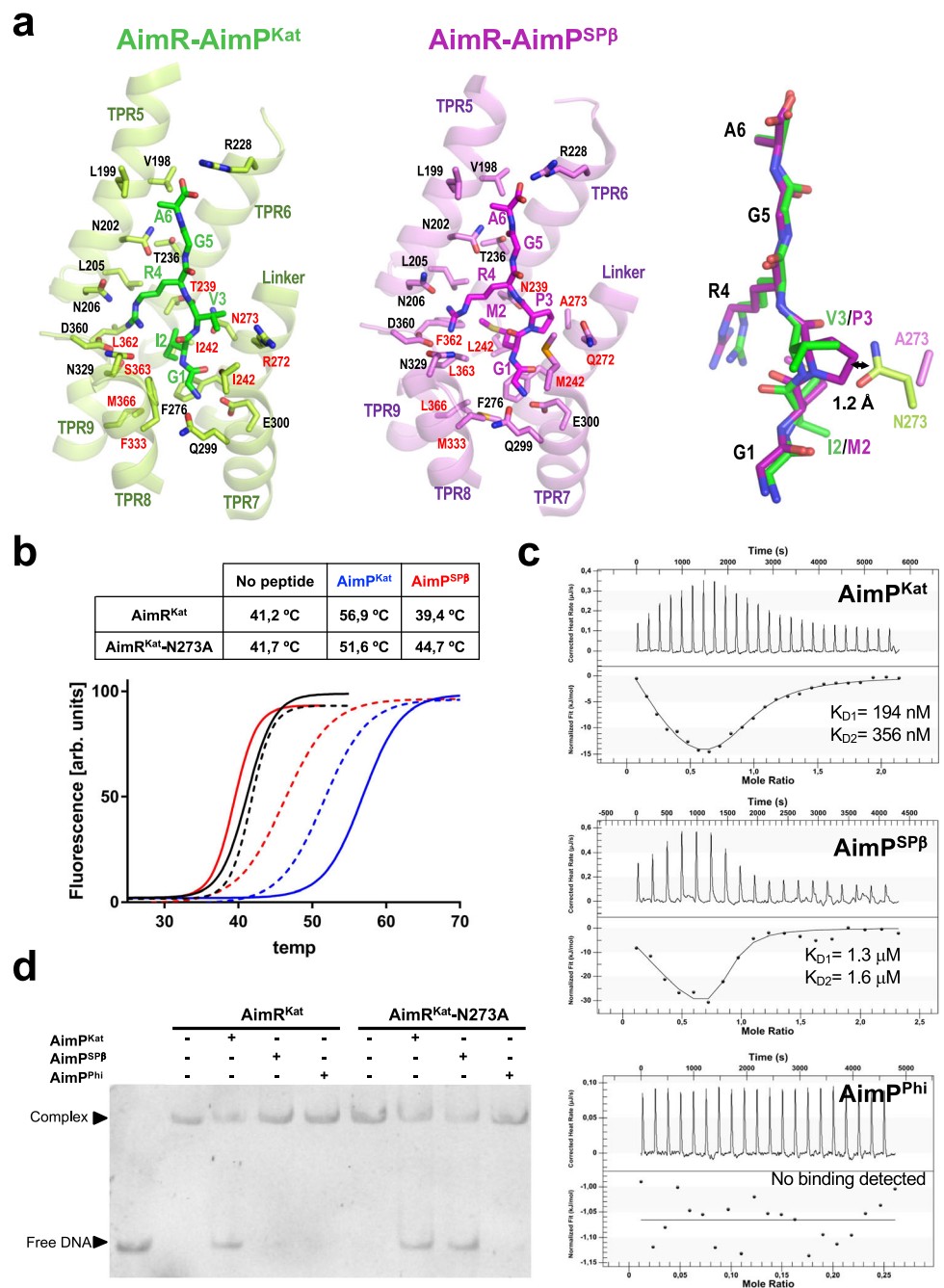

**Fig. 7 AimR peptide selectivity. a** Close view of AimPs bound at the corresponding binding sites AimR$^{Kat}$ (*left*) and AimR$^{SP\beta}$ (*centre*) showing the AimPs and the AimR interacting residues in sticks. The AimR structural elements where the recognition residues are placed are shown in translucent cartoon and labelled. Conserved and variable interacting residues between both AimRs are labelled in black and red, respectively. In the *right*, the superimposition of both AimPs is shown in stick as well the side-chain for the residue in position 273. The distance between the side chains of AimR$^{Kat}$ N273 and AimP$^{SP\beta}$ P3 is indicated. **b–d** Mutation of AimR$^{Kat}$ residue 273 alter AimP sensitivity. In (**b**) thermal unfolding curves of AimR$^{Kat}$ wt (solid lines) and N273A mutant (broken lines) alone (black) or in presence of AimP$^{Kat}$ (blue) and AimP$^{SP\beta}$ (red) are shown. The unfolding Tms showed in the table support AimP sensitivity variation induced by the mutation. **c** ITC measurement of AimR$^{Kat}$-N273A binding affinity for AimP$^{Kat}$, AimP$^{SP\beta}$ and AimP$^{Phi}$. In case of binding, thermogram were adjusted to two-binding site model and the two $K_D$ ($K_{D1}$ and $K_{D2}$) values, one per monomer in the AimR dimer, are shown. In (**d**) EMSA analysis shows that both AimP$^{Kat}$ and AimP$^{SP\beta}$ peptides induce DNA releases for the N273A mutant form of AimR$^{Kat}$ but only AimP$^{Kat}$ for the wt form. Both AimR$^{Kat}$ wt and N273A mutant are insensitive to AimP$^{Phi}$. EMSA assays have been repeated independently three times with similar results. Source data are provided as a Source Data file.

*subtilis* or other *Bacillus* species (Supplementary Fig. 10a). Sequence analyses of four such homologues for each of the AimRs showed sequence identity for the DBD (residues 1 to 49) of ~90% (only four conservative changes and one non-conservative), with 100% identity for the DNA recognition α3 helix (Supplementary Fig. 10b). The 90% identity extends beyond the DBD to encompass TPRs 1 and 2. Consistent with the identity in α3 helices and the almost identical DBDs, each AimR group has operators with identical palindromic sequences (only a single change in an operator of AimR$^{SP\beta}$ group at a non relevant

position for recognition[8]) separated by the prototypical 25 bp spacer (Supplementary Fig. 10c). Compatible with their function, these operators are located downstream of *aim*P genes and precede the putative *aim*X. In contrast, the TPR domain (TPR3-9 corresponding to residues 100 to C-terminal) exhibits a much lower (<40%) sequence identity (Supplementary Fig. 10b), suggesting that they recognise different peptides. We confirmed this by identifying the ORFs encoding the putative AimPs. In support of our previous results, the mature AimPs presented variations in positions 2 and 3 (Supplementary Fig. 10d), confirming alternative peptide specificity for their cognate AimR receptor. Therefore, our results support AimR modularity and suggest the existence of a putative mechanism of cross regulation where different AimRs recognise identical operators in the phage genomes but are controlled by different peptides.

## Discussion

The recent description of the arbitrium communication system was a game changer in the perception of phage ecology[16]. Although arbitrium was reminiscent of the RRNPP family of QS systems, which is characteristic for the bacteria hosting the phages communicating via this system, it was initially dismissed as a member of this family[2]. Therefore, in order to understand its mechanism of action, a molecular dissection of the system was mandatory. In this context, it was not surprising that, almost simultaneously, the structures of AimR from the SPβ phage in its apo and peptide-bound states were reported by three different laboratories. Interestingly, while the reported structures for the individual protomers were almost identical, the conformation of the dimers was not the same and therefore, the regulatory mechanisms deduced from these structures were completely different. The structures presented by some groups showed that AimP binding to AimR did not induce structural changes in the receptor. In this conformation, the DNA recognition helices are positioned extremely far apart from each other (around 100 Å). The fact that these structures presented an unusual DBD separation even in the apo form prompted the authors to propose that in the absence of peptide, the DNA would induce a conformational change that would approach the DBDs to a more canonical separation. Since in the presence of the AimP peptide these authors did not see any change in the AimR structure, they hypothesised that the peptide would block the DNA-induced conformational change in AimR, preventing DNA binding[9,11]. In contrast, the structures solved by our group showed high plasticity in the apo form of AimR and that AimP induces, as in other members of the RRNPP family[2], major conformational changes that cause a more compact and "closed" conformation in AimR. Further, our study revealed that the AimR^SPβ box presents an unusual DNA operator composed of two palindromic 6 bp repeats separated by a long 25 bp spacer[8]. Therefore, the atypical DBD separation observed in the AimR apo form is necessary to accommodate the non-canonical DNA operator without any modification to the AimR structure. This was confirmed after solving the structure of the AimR^SPβ-DNA complex[8]. In this way, the "closed" conformation observed in the structure of AimR^SPβ bound to the regulatory AimP explains how the conformational changes to the DBDs prevent the DNA binding. It should be noted that this plasticity of the apo form and the atypical separation of the DBDs is biologically relevant, as our assays showed that AimR^SPβ binds alternative operators to *aim*X in the phage genome whose boxes are spaced by more than 28 bp[8].

In this manuscript, we have not only confirmed and clarified the mechanism of action by which AimP blocks *aim*X expression in the SPbeta phages, but also have explained the origin of these discrepancies. While all our structures were obtained after removing the His-tag present in the AimR proteins, this was not the case for the other structures, which were obtained using AimRs that retained the His-tag in their C-terminal regions. Unfortunately, the presence of this extra C-terminal tag altered the AimR conformation obtained in the crystals. In fact, the extra residues L387, E388, H389 and H390 are clearly defined in the experimental map, being embedded in the dimer interface. In this position, the C-terminal tag residues participate in crucial interactions with TPR3 and TPR4, pulling away from this second dimerisation interface, which works as a slipping surface for receptor plasticity. This case is only one example of how the introduction of additional elements (His-tag residues) can distort results and emphasises the need for caution in their use when attempting to understand mechanistic inferences. Several reported examples have shown that the introduction of even a small purification tag may modify the protein behaviour, altering (increasing or decreasing) its intrinsic activity or conferring new capabilities[17–20]. In general, it has been observed that tags do not usually have major structural impacts[21]. However, this is not true in some cases and some structures have shown how the tags establish multiple interactions[22,23], inducing conformational changes that, in the absence of other structural data, may lead to erroneous mechanistical interpretations, exemplified by some of the published AimR structures. Moreover, the structural characterisation of the AimR of phage phi3T was carried out in parallel with that of phage SPβ showing striking differences. For AimR^Phi, a dimer-monomer transition induced by the AimP was observed, proposing different mechanisms of action for the inhibitory peptide. However, these structural analyses were performed with a double mutant of AimR^Phi that, precisely, involved two residues located in the dimerisation region[9]. The recent biophysical characterisation of AimR^Phi wt has shown that the inhibitory peptide does not induce monomerisation of the receptor, and that the receptor behaves as a dimer in the presence and absence of AimP[13], supporting a mechanism of action similar to that observed for AimR^SPβ and AimR^Kat. In parallel, these recent observations with AimR^Phi confirm that the use of modified variants (by tags or mutations) can have undesirable parallel functional impact and that it is necessary to be cautious about the biological mechanism deduced.

The confirmation of AimRs as receptors with a high degree of plasticity, whose intrinsic flexibility is restricted by the peptide, has important biological and evolutionary implications. Here, we have confirmed the AimR modularity with DBD effector (N-terminal) and TPR regulatory (C-terminal) portions that, interestingly, seem to show different evolutionary pressures. Identical DNA operators are recognised by AimRs that are regulated by different, but related, peptides. Given that the plasticity of AimR allows for the recognition of alternative operators with different spacing[8], the conservation of DBDs would suggest that AimR could be controlling similar and essential processes for the phage life cycle beyond *aim*X. This could lead to a scenario where the presence of two related phages infecting the same host activate similar pathways, even though they do not communicate directly with each other. Therefore, the arbitrium system could allow the development of social behaviours such as altruism or cooperation, which have already been observed for other viruses[24]. Furthermore, the structures of AimR^Kat confirm that in the presence of the peptide, the DNA recognition helices remain exposed in a competent DNA-binding disposition, as was also observed for AimR^SPβ[8]. However, the relative distance of the helices in the peptide-stabilised state is different in each receptor, which indicates that in this conformation the AimR could regulate different operators with alternative spacing between the boxes. This attractive proposition, which is currently under analysis, would open the door to a dual regulation role for arbitrium, one general

for "relative" phages and one species-specific modulated by the cognate AimP.

The comparative analyses of AimR$^{Kat}$ and AimR$^{SP\beta}$ structures revealed that a limited number of residues provide the peptide specificity in related AimRs. By changing only one of these residues we modified the receptor specificity, making the AimR responsive to a non-cognate AimP. Similarly, a single residue mutation in RapF, another member of the RRNPP family, caused a change in peptide specificity[21]. These results would support the hypothesis that phages can modify their affinity and selectivity for regulatory peptides with minimal mutational changes in their AimRs. In this way, phages would be rapidly segregated during their evolution so that, as proposed, evolved phages would only communicate with their own progeny[4]. However, the results also indicate that these minimal changes do not result in complete isolation, as the receptors can still "hear", albeit with different intensity, peptides from related phages. Therefore, in this evolutionary process, the receptors could exhibit crosstalk capabilities allowing communication between different phages. This possibility has not been evaluated experimentally as studies of arbitrium systems have been reduced to a very limited number of cases and, moreover, belonging to very distinct groups of phages. Our vision, which is entirely speculative, is that related phages may not only present cross regulation but also crosstalk, showing the arbitrium system to be involved in complex phage-phage interactions. The confirmation of this hypothesis would open the interesting possibility that this quorum-sensing mechanism could confer social behaviours to arbitrium-carrying phages.

## Methods

**Bacterial strains and growth conditions.** Bacterial strains used in this study are listed in Supplementary Table 2. *B. subtilis* strains 168 and Δ6 were obtained from the *Bacillus* Genetic Stock Centre (BGSC) and *B. subtilis* subsp. KATMIRA1933 was a gift from professor Wilfried J.J Meijer. *B. subtilis* strains were grown at 37 °C on LB (Miller) agar plates or in LB (Miller) broth liquid medium shaking at 210 rpm. *Escherichia coli* DH5α was grown at 37 °C on LB (Lennox) agar plates or in LB (Lennox) broth shaking at 210 rpm. When required, antibiotics were utilised at the following concentrations: kanamycin (10 μg ml$^{-1}$), ampicillin (100 μg ml$^{-1}$) or spectinomycin (100 μg ml$^{-1}$).

**Plasmids and cloning.** Plasmids generated in this study are listed in Supplementary Table 2 and primers used are listed in Supplementary Table 3. The AimR$^{SP\beta}$ and AimR$^{Kat}$ were cloned into the *amyE* integration vector pDR110 under the control of the IPTG inducible promoter $P_{spank}$[25]. Cloning was performed after PCR amplification of the appropriate template DNA with the same forward primer (AimR-SPβ-1mH) but different reverse primer (AimR$^{SP\beta}$: AimR-SPβ-2cS; and AimR$^{Kat}$: AimR-Katmira-1cS). Competent cells preparation and transformation was performed as described by Bron et al.[26]. Briefly *B. subtilis* cells were grown in minimum medium to early stationary phase to induce natural competence and 1 μg of plasmid was added and incubated at 37 °C for 20 min with shaking at 210 rpm. Then, 50 μl of 10% casamino acids were added to the cells and incubated for a further 90 min with the same conditions. The culture was centrifuged at 8000 × g for 5 min, 800 μl of the supernatant removed, and the pellet re-suspended and plated out onto the relevant antibiotic plates. Plates were incubated at 37 °C for 24 h.

For AimR$^{Kat}$ protein expression, the Katmira *aim*R gene was amplified using primers Kat33Plic_FW and Kat33Plic_RV and genomic DNA from *Bacillus subtilis* subsp. KATMIRA1933 as template. The PCR product was purified and cloned into pLicSGC1 plasmid using Ligation-Independent Cloning (LIC) system as previously described[27]. The resulting pLIC-AimRKat33 plasmid expressed AimR$^{Kat}$ (residues 1–386) with an N-terminal 6xHistag followed by a TEV protease cleaving site. The AimR$^{Kat}$-N273A mutant was generated using Q5 Mutagenesis Kit (NEB), pLIC-AimRKat33 plasmid as template and primers Kat33$^{N273A}$_FW and Kat33$^{N273A}$_RV. For production of AimR$^{SP\beta}$ with the C-terminal His-tag, the *aim*R gene from the SPβ phage was amplified using primers SPbCter_FW and SPbC-ter_RV and genomic DNA from *B. subtilis* strain 168 as template. In parallel, pET21b vector (Novagen) was amplified using primers pet21_FW and pet21_RV. Both PCR products were purified and assembled using NEBuilder kit (NEB). The resulting plasmid expressed AimR$^{SP\beta}$ (residues 1–386) plus LEYAHHHHH C-terminal tag (AimR$^{SP\beta}$-II).

**Bacteriophage induction assay.** For induction, an overnight culture was diluted 1/100 in LB media supplemented with 0.1 mM MnCl2 and 5 mM MgCl2 and then grown at 37 °C with 210 rpm shaking until reaching absorbance 0.2 at 600 nm. This step was repeated twice to ensure the cells were in exponential growth. After the second growth, Mitomycin C (MC) at 0.5 μg ml$^{-1}$ was added to the culture. Where experiments were performed to test the effect of peptide on phage titre, 5 μM of the relevant peptide was added before starting the second growth to ensure the bacteria cells were able to incorporate the peptide. The induced cultures were incubated at 30 °C with 80 rpm shaking for 4 h and then left overnight at room temperature. Following lysis, samples were filtered using 0.2 μm filters and lysates were stored at 4 °C until use.

**Bacteriophage titering assay.** The number of phage particles contained in the phage lysate of interest was quantified by a titering assay. An overnight culture of the relevant recipient strain (normally *B. subtilis* Δ6 or with the corresponding plasmid) was diluted 1/100 in LB supplemented with 0.1 mM MnCl2 and 5 mM MgCl2 and then grown at 37 °C with 210 rpm shaking until reaching absorbance 0.2 at 600 nm. When required 1 mM IPTG was added. Then, 100 μl of recipient bacteria was infected with 100 μl of serial dilutions of phage lysate in phage buffer (PhB; 1 mM NaCl, 0.05 M Tris pH 7.8, 0.1 mM MnCl2, 5 mM MgCl2) at room temperature for 10 min and 3 ml of phage top agar (LB media supplemented with 0.1 mM MnCl2 and 5 mM MgCl2 and 0.7 % agar) at 55 °C was added to the culture-phage mix and immediately poured over phage base agar plates (LB media supplemented with 0.1 mM MnCl2 and 5 mM MgCl2 and 1.5% agar). Plaques were counted after overnight growth at 37 °C temperature and photographed.

**Recombinant protein expression and purification.** AimR$^{Kat}$ wt and mutants were produced and purified using a similar protocol to the previously described for AimR$^{SP\beta}$ [8]. Briefly, a flask with 50 ml of LB medium supplemented with 100 μg ml$^{-1}$ of ampicillin and 33 μg ml$^{-1}$ of chloramphenicol was inoculated with a single colony of *E. coli* strain BL2_Codon plus (DE3) RIL (Agilent) carrying the corresponding expression plasmid (Supplementary Table 2) and grown overnight at 37 °C. The culture was used to inoculate 4 L of LB medium supplemented with 100 μg ml$^{-1}$ ampicillin and 33 μg ml$^{-1}$ chloramphenicol and was incubated at 37 °C with shaking at 190 rpm. Cell growth was monitored until absorbance reached 0.4 at 600 nm. Then temperature was reduced to 20 °C and protein expression was induced by adding IPTG to a final concentration of 0.2 mM. After 16 h, cells were pelleted by centrifugation at 4500 × g for 45 min and the pellet was stored at −80 °C. For protein purification, the pellet was suspended in lysis buffer (25 mM Tris-HCl pH 8, 250 mM NaCl) and disrupted by sonication on ice. The sonicated lysate was clarified by centrifugation at 10,000 × g for 1 h. The supernatant was loaded onto a 5 ml HisTrap FF (GE Healthcare), washed with lysis buffer and eluted with lysis buffer supplemented with 500 mM imidazole. In order to remove the His-tag, AimR$^{Kat}$ was digested with TEV protease (50:1 molar ratio protein:TEV) and dialysed against dialysis buffer (25 mM Tris-HCl pH8, 250 mM NaCl, 1 mM 2-βmercaptoethanol and 0.5 mM EDTA). The sample was concentrated in a centrifugal filter (Amicon ultra 30 KDa) and loaded onto a Hi-Load Superdex 200 16/60 (GE Healthcare) gel filtration column previously equilibrated with lysis buffer. Fractions containing the purest protein were pooled, concentrated at 90 mg ml$^{-1}$ and stored at −80 °C. Typical yields were 25 mg recombinant protein/L of culture medium. AimR$^{Kat}$-N273A was purified following the same protocol. AimR$^{SP\beta}$ was produced and purified as previously described[8]. AimR$^{SP\beta}$-II was purified using the same protocol, but excluding the TEV digestion step.

**Protein crystallisation and data collection.** The crystals were grown in hanging drops at 21 °C with a vapour-diffusion approach. Initial crystallisation trials were set up in the Cristalogenesis service of the IBV-CSIC using commercial screens JBS I, JBS II (JENA Biosciences) and MIDAS (Molecular Dimensions) in 96-well plates (Swissci MRC2) using equal volumes of protein at 10 mg/ml in lysis buffer and precipitant. The apo form of AimR$^{Kat}$ crystallised in 0.1 M Lithium sulfate, 0.1 M HEPES 7.0 and 30% w/v Polyvinylpyrrolidone. Crystals of AimR$^{Kat}$ in complex with GIVRGA (AimR-AimP$^{kat}$) peptide were obtained adding 1 mM of the peptide to the protein solution and using 0.2 M Potassium acetate, 0.1 M MES 6.0, 15% v/v Pentaerythritol ethoxylate (15/4 EO/OH) and 3% v/v Jeffamine T-403 as precipitant. Crystals of AimR$^{Kat}$ in complex with DNA were obtained using protein at 10 mg/ml mixed with duplex DNA (IDT) at final molar stoichiometry 2:1 protein:DNA and 50% PEG400, 0.1 M Sodium acetate and 0.2 M Lithium sulphate as precipitant. Crystals grew in 1–7 days and were directly flash frozen in liquid nitrogen. Diffraction data was collected from single crystals at 100 °K at XALOC beamline (ALBA Synchrotron), I24 beamline from DLS (Diamond Light Source) and ID23-2 from ESRF (European Synchrotron Radiation Facility). Data sets were processed with XDS[28] and reduced using Scala[29](CCP4). The data-collection statistics for data sets used in structure determination are shown in Supplementary Table 1.

**Phase determination, model building and refinement.** The structures reported in the manuscript were solved by molecular replacement using Phaser[30] (CCP4). To solve AimR$^{Kat}$ structure the PDB 6HP3 was used as search model in the molecular replacement. Then, the AimR$^{Kat}$ refined model was used to AimR-AimP$^{Kat}$ structure. Finally, the structure of AimR$^{Kat}$-DNA complex was solved using PDB 6HP7 as search model. To generate the final models, several rounds of manual model building were completed using COOT[31] and computational refinement with Refmac[32] (CCP4). Refinement statistics are summarised in Supplementary Table 1. Protein

assemblies and interactions analysis were carried out with PISA and CONTACT (CCP4). Figures for three-dimensional structures were generated with Pymol (version Open-source PyMOL 1.8.x. https://pymol.org/2/).

**Thermal shift assay**. The thermal shift assay was conducted in a 7500 Fast Real time PCR System (Applied Biosystems) as previously described[33]. Briefly, samples of 20 μl buffer (20 mM Tris pH 8 and 250 mM NaCl) containing 5× Sypro Orange (Sigma-Aldrich) and 20 μM of protein were loaded in 96-well PCR plates. To calculate the Tm in the presence of the peptide, 0.5 μM of GMPRGA or GIVRGA peptide was added to the mixture. Samples were heated from 25 to 85 °C in steps of one degree. Fluorescent intensity was plotted versus temperature and integrated with GraphPad Prism software using a Boltzmann model to calculate melting temperatures.

**EMSA assays**. AimR binding to its operator and the inhibition induced by the arbitrium peptide was analysed by native polyacrylamide and agarose gel electrophoresis. Double strand DNA primer probes were purchased from Macrogen. DNA (10 ng μl$^{-1}$) and AimR (from 0.5 μM) protein was mixed in EMSA buffer (50 mM tris pH8, 250 mM NaCl). The samples were incubated for 10 min at room temperature. For peptide inhibition assay, protein was preincubated with 0.5 μM peptide for 10 min before DNA addition. Electrophoresis was then performed in 8% polyacrylamide gels in Tris-Borate-EDTA (TBE) buffer for about 150 min at 100 V at 4 °C.

**Isothermal titration calorimetry (ITC)**. ITC assays were carried out in a Nano ITC Low Volume (TA instruments). In the assays, proteins (AimR$^{Kat}$ or AimR$^{Kat}$-N273A) and peptides concentration were 10 and 50 μM, respectively, and both were diluted in buffer 25 mM Tris-HCl pH 8, 250 mM NaCl. The experiments were performed at 25 °C. The data obtained was integrated, corrected and analysed using the NanoAnalyze software (TA Instruments) with a single-site or two-site binding models[34].

**Biolayer interferometry (BLI)**. The Binding affinity (K$_D$), association (k$_{on}$) and dissociation (k$_{off}$) rate constants between AimRs and DNA were measured by biolayer interferometry (BLI) using the BLITz system (FortéBio). Biotinilated DNA probes (IDT) (Supplementary Table 3) were immobilised in Streptavidin biosensors (FortéBio) at 50 μg ml$^{-1}$. Biosensor hydration, baselines and dissociation analysis were carried out in phosphate buffer (PBS) supplemented with 0.01% tween and 0.1% BSA. Proteins were diluted in PBS. At least four different dilutions of AimR were used in a range between 0.015 and 2.5 μM. A blank using a chip without bounded DNA was used to subtract unspecific binding. Kinetic value calculations and data analysis were performed using BLItz Pro 1.2 software employing a 1:1 model to fit the data.

**Size exclusion chromatography with multi-angle light scattering (SEC-MALS)**. SEC-MALS experiments were performed using a Wyatt DAWN HELEOS-II MALS instrument and a Wyatt Optilab rEX differential refractometer (Wyatt) coupled to an AKTA pure system (GE Heralthcare)[35]. 50 μl of protein at 5 mg ml$^{-1}$ were injected in a KW-803 (Shodex) column equilibrated with 25 mM Tris pH 8, 250 mM NaCl. When testing the peptide in the AimR$^{Kat}$ oligomeric state, 1 mM peptide was added to the injected sample and the running buffer was supplemented with peptide at a final concentration of 1 μM. The Astra 7.1.2 software from the manufacturer (Wyatt) was used for acquisition and analysis of the data.

**Peptides**. Peptides used in this study were purchased from Proteogenix at 95% purity.

**Reporting summary**. Further information on research design is available in the Nature Research Reporting Summary linked to this article.

## Data availability

Coordinates and structure factors for AimR$^{Kat}$, AimR-AimP$^{Kat}$ and AimR$^{Kat}$-DNA have been deposited in the Protein Data Bank under access codes 6S7I, 6S7L and 7Q0N, respectively. All relevant accession codes and identifiers are provided within the paper. Source data are provided with this paper.

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

## Acknowledgements

We would like to thank the IBV-CSIC Crystallogenesis Facility for protein crystallisation screenings as well as Dr. Wilfried Meijer from CBMSO-CSIC (Spain) for provision of *B. subtillis* strain. The structural results reported in this article derive from measurements made at the synchrotron DLS (Didcot, UK), ALBA (Cerdanyola del Valles, Spain) and ESRF (Grenoble, France). Data-collection experiments for the best crystals were carried at XALOC, I24 and ID23-2 beamlines at ALBA, DLS and ESRF Synchrotrons, respectively. X-ray diffraction data collection was supported by block allocation group (BAG) DLS Proposal MX28394, ALBA Proposals 2020074406 and 2021075216, and ESRF proposal MX-2351. We acknowledge the ESRF, ALBA and DLS synchrotrons for provision of beam time and we would like to thank beamline staff for assistance. This work was supported by grants BIO2016-78571-P and PID2019-108541GB-I00 from Spanish Government (Ministerio de Economía y Competitividad y Ministerio de Ciencia e Innovación) and PROMETEO/2020/012 by Valencian Government to A.M, and grant MR/M003876/1, MR/V000772/1 and MR/S00940X/1 from the Medical Research Council (UK), BB/N002873/1, BB/V002376/1 and BB/S003835/1 from the Biotechnology and Biological Sciences Research Council (BBSRC, UK), ERC-ADG-2014 Proposal nº 670932 Dut-signal (from EU), and Wellcome Trust 201531/Z/16/Z to J.R.P.

## Author contributions

A.M., F.GdS and JRP designed the study. F.GdS., N.Q-P., A.B. and A.M performed experiments. A.M., F.GdS., N.Q-P., A.B and J.R.P wrote the paper. A.M and J.R.P. supervised the project and acquire funding.

## Competing interests

The authors declare no competing interests.
