## [Peer Review File · Nature Communications]

Reviewer comments, first round of review

Reviewer #1 (Remarks to the Author):

The arbitrium system is peptide-based communication system to coordinate the lysis-lysogenic cycle of phage infecting bacteria. So far, the system or the system-like had been found in diverse bacteria. The mechanism underlying regulations were reported recently based on functional and structural analyses. However, many questions about the regulatory mechanism are still to be addressed. In this study, Francisca et al report a homolog protein of AimRSPb named AimRkat, which share approximate 70% amino acid identity, particularly the DBDs is almost identical, and the fragment AA137-386 also share 50% identity. Moreover, they reported the two homologs could recognize reciprocal target DNA sequence. Then the author assumed this phenomenon as crossregulation, which sounds interesting. However, the crystal structures of AimRkat (apo and AimP-bound) which are almost identical to the structures of AimRSPBeta, together with in vitro and in vivo current experiments could not substantially support the crossregulation conclusion. In addition, the manuscript is rather rough, it would benefit from careful proofreading as there are quite a few errors in text and pictures. Therefore, I suggest that all questions have to be addressed as followed, it should help to improve the quality of the manuscript.

Comments as following:

1. Line 166-167 : The arbitrium system of Katmira phage localised to clade 2 of clustered arbitrium systems during previous phylogenetic analyses. Needing correct.
2. Line 173-174 : We therefore scrutinised this region in the 174 Katmira phage genome to identify the DNA binding region recognised by AimRkat. Correct.
3. Line 181-184 : Making clarity and correct error. For example: the above (not previous) results shown (not suggest) that ---could bind to, not will. ----- . Correct this sentence
4. Line 187: To test whether AimRkat and AimRSPb binds to identical AimR boxes. AimR box is AimX box? Correct.
5. Line 190-191: The identical affinities reported in the manuscript, please define the binding constant by ITC, or SPR, or MST.
6. Line 193: did not revealed should be corrected into: did not reveal; “confirming the specificity of the observed interactions” suggest to change: “confirming the specificity of the observed interactions between -----“.
7. Line 196-197: Rewrite this sentence and add the Fig. 1e.
8. Line 231-233 : The current finding could solve the discrepancies about the mechanism of action for these receptors? IMO, it still needs more work to further understand the discrepancies of action mechanism. This sentence seems overstated, it will lead to misunderstanding.
9. Line 246-247 : Structural comparison showed please in supplementary data, to know similar or identical? If difference, please clarify and interpret.

10. Line 252-255 : The crystals of two of these structures, PDBs 6IPX and 6JGS, show the same space group (P21212) and almost identical unit cell, and were obtained in similar crystallisation conditions (Supplementary Fig. 2). Three questions of this sentence: 1. Where to find 6JGS? 2. which crystals crystallization conditions is similar to 6IPX? 3. Which are the same space groups and the same conditions ? Please clarify

11. Line 256: 6JGS. Correct

12. Line 265-273: There are not sufficient evidences to support this explanation. I went through the reference paper, and present my understanding below: AimRKat and AimRSPb-I apo structures are similar to structure of apo phAimR in close state, but both of them show conformational changes when bound to cognate ligands. So here the current structure of AimRKat is probably one state of the serially dynamical conformations, and as well as AimRSPb-I and AimRSPb-II.

13. Line 359: The C-terminal His tag disturbs dimer interface in AimR. No sufficient evidences prove the conclusion. Explaining why it could rescue in vivo using the C-terminal 6×His-tag in previous report.

14. It should highlight that the structure of phAimRY341A/E371A-AimP in monomer, not wild type, was a double mutant variant, the presentation of the current manuscript would lead to misunderstanding.

15. In the manuscript, several paragraphs discussed the effect of the C-terminal tag of AimRspBeta, but never talk about the dynamics of AimRph, and never discuss about the effect of the C-terminal tag of AimRph. The author even considerate that SAIRGA could disrupt the dimeric state of AimRph. Structures of these receptors or bound to ligands are probably different snapshots, which have been indicated by gel-filtration assays. The authors should clarify these different structures from the same spBeta phage group, why are same, why not.

16. One more structures should be solved of the AimR superfamily, the spBeta subfamily is the small part. In particularly, the identity of these protein would be better if lower 30%. Thus, it will provide insights into the differences between AimRKat and AimRSpB and AimpPh, and other receptors.

17. In Fig. 1b The peptide sequences of AimpSpB and AimpKat are wrong.

18. In Fig. 1c the DNA Sequence name should be AimXSpB or Kat

19. In Fig. 1d, the DNA operator of phi3T indicate have a bit interaction with AimRKat, clarify please.

20. In Fig. 1e or legend, the Kd has to be indicated.

21. In Fig. 2a. Error bar is approximate zero, interpret please.

22. In Fig. 2b, the parallel experiments in curves as supplementary data or main figs are needed.

23. Fig. 5b, label or legends represent clearly. Through all text, authors highlight that the regulation mechanism of AimRKat is most similar to AimRSpB, and their structures should be identical. To support this point, structure of AimRKat-AimXkat is requested, then overall structural comparisons between apo-AimRSpB and AimRKat or bound to DNA ligands are needed.

24. In Fig 6. C, the result shows AimpSpB could induce DNA release as well as Aimpkat, it essentially needs in vivo experiments (knock-out of AimRSpB or Aimpkat of corresponding phage stains, respectively, then perform complement experiments using cross receptors) to further confirm this

conclusion, moreover, the measurement of binding affinity for AimRKAT-N273A and AimpSpB is also essentially needed. More experiments and discussion are needed to substantiate the current point.

25. More dynamical experiments could further exemplify the conclusion of this manuscript.

26. In this work, the author characterized the structure of AimRKat, which is almost identical to AimRSpB, However, many structures have been reported before (References 6-9), the author has cited them, it will be important to objectively point out the difference and similarity based on in vivo, in vitro functional experiments and more structures among the findings in current work and the previous reports to avoid overstating and misunderstanding.

Reviewer #2 (Remarks to the Author):

Professor Alberto Marina and the co-authors determined the structure of apo AimRkat and the complex with the peptide (GIVRGA) in this manuscript. However, they and other groups solved the structure of AimRbeta-AimP and AimRPhi-AimP almost simultaneously in 2019. The structural basis of how AimR recognized AimP and how AimP prevented AimR binds to AimR are clear. Although the author noted that a little difference between their results and they further proved their hypothesis in this study that the AimRKat with a chimeric trait, which is further supports their point. But I believe the novelty is not enough for the publish in Nature Communications. I suggest that they can consider for transferring it to Communication Biology.

I also have two questions:

1. I am interested that if does it impact the ability of AimR binds to AimX if the 6XHis tag in the C terminus? Because the structure of AimR-AimX(PDB: 6GJ8) was also solved. Normally, if the transcription factors are not in the right condition, it can not bind the promoter, so please compare the affinity between AimR-6XHis and AimR alone with AimX.

2. A blank space should be between the number and their units such as 150 mM, there are too many little errors in this manuscript, please check it.

REVIEWER COMMENTS:

Reviewer #1 (Remarks to the Author):

The arbitrium system is peptide-based communication system to coordinate the lysis-lysogenic cycle of phage infecting bacteria. So far, the system or the system-like had been found in diverse bacteria. The mechanism underlying regulations were reported recently based on functional and structural analyses. However, many questions about the regulatory mechanism are still to be addressed. In this study, Francisca et al report a homolog protein of AimRSPb named AimRkat, which share approximate 70% amino acid identity, particularly the DBDs is almost identical, and the fragment AA137-386 also share 50% identity. Moreover, they reported the two homologs could recognize reciprocal target DNA sequence. Then the author assumed this phenomenon as crossregulation, which sounds interesting. However, the crystal structures of AimRkat (apo and AimP-bound) which are almost identical to the structures of AimRSPbeta, together with in vitro and in vivo current experiments could not substantially support the crossregulation conclusion. In addition, the manuscript is rather rough, it would benefit from careful proofreading as there are quite a few errors in text and pictures. Therefore, I suggest that all questions have to be addressed as followed, it should help to improve the quality of the manuscript.

Comments as following:

1. Line 166-167: The arbitrium system of Katmira phage localised to clade 2 of clustered arbitrium systems during previous phylogenetic analyses. Needing correct.

Corrected.

2. Line 173-174: We therefore scrutinised this region in the 174 Katmira phage genome to identify the DNA binding region recognised by AimRKat.

Corrected.

3. Line 181-184: Making clarity and correct error. For example: the above (not previous) results shown (not suggest) that ---could bind to, not will.

Thanks for the suggestion. The sentence has been corrected and is now clearer.

4. Line 187: To test whether AimRKat and AimRSPb binds to identical AimR boxes. AimR box is AimX box? Correct.

We thank the reviewer for his/her comment, which helped us to clarify this point. In the manuscript we referred to the AimR box as the operator located before *aimX* that is recognised by AimR (the two 6bp inverted repeats separated by 25 bp), as is described in our previous work (doi: [10.1016/j.molcel.2019.01.025](https://doi.org/10.1016/j.molcel.2019.01.025)). To make this point clear in the new version of the manuscript, we have referred to this DNA sequence as *aimX*-box^{SPβ} (ATCACTTAAATATTAGGTTTTAATAACATCTAGTGAT). Analogously, in the new version of the manuscript, we refer to the palindromic DNA sequence upstream from *aimX* in the Katmira phage that is recognised by AimR^{kat} as *aimX*-box^{kat}

(ATCACTTAAATATTAAGTTTTTATAACATCTAGTGAT). Although it has been shown that AimR^{SPβ} can bind to other operators present in the SPβ phage genome (showing degenerated *aimX*-box^{SPβ} sequences (doi: [10.1016/j.molcel.2019.01.025](https://doi.org/10.1016/j.molcel.2019.01.025))), in the present manuscript we will refer to *aimX*-boxes as the biologically confirmed AimR binding boxes preceding the *aimX* genes. This clarification has been introduced in the section “*AimR^{Kat} and AimR^{SPβ} recognise identical operators but are regulated by different peptides*”.

5. Line 190-191: The identical affinities reported in the manuscript, please define the binding constant by ITC, or SPR, or MST.

Upon the Reviewer's request, we have characterised by Biolayer interferometry (BLI) the affinities of the AimRs for their DNA boxes. These results have been now included in the manuscript in section “*AimR^{Kat} and AimR^{SPβ} recognise identical operators but are regulated by different peptides*” and are also summarised in (new) Table 1. The results obtained allow us to confirm that both receptors, AimR^{Kat} and AimR^{SPβ}, bind to *aimX*-box^{SPβ} and *aimX*-box^{Kat} with almost identical affinities, being the affinities (K_D) for both boxes and receptors ranging 1.93-2.3x10⁻⁸ M .

The Reviewer's comment also provided to us the opportunity to measure in parallel the affinity for the DNA boxes of the AimR^{SPβ} including the C-terminal His-tag, which our structural analysis proposes as responsible for the conformational changes observed in AimR^{SPβ}-II structures. The BLI analysis shows that the presence of the C-terminal His-tag decreases the affinity of AimR^{SPβ} for *aimX*-box^{SPβ} and *aimX*-box^{Kat} around one order of magnitude (from 19 nM to 116 nM) Comparison of the interaction kinetics of AimR^{SPβ} with and without the C-terminal His-tag shows that the differences in affinity are due only to a reduction in the K_{on} (from 3.8-4 x10⁶ to 6.3-6.8 x10⁵ M⁻¹ s⁻¹), with the K_{off} remaining the same for both receptors (7.3-7.8 10⁻² s⁻¹). These results confirm the hypothesis that the presence of the His-tag hinders the conformational changes required for DNA binding upon contact with the dimerising surface of AimR. However, once the protein has bound to DNA, which involves expelling these His-tags from the dimerisation surface as observed in PDB 6JG8, the protein acquires the biologically competent closed conformation. Therefore, the newly incorporated data presented in Table 1 confirm and support the original conclusions, making the manuscript more solid.

6. Line 193: did not revealed should be corrected into: did not reveal; “confirming the specificity of the observed interactions” suggest to change: “confirming the specificity of the observed interactions between ----”.

Corrected.

7. Line 196-197: Rewrite this sentence and add the Fig. 1e.

The sentence has been rewritten (now lines 226-229) and a new Fig. 1e (ITC experiment) has been added.

8. Line 231-233; The current finding could solve the discrepancies about the mechanism of action for these receptors? IMO, it still needs more work to further

understand the discrepancies of action mechanism. This sentence seems overstated, it will lead to misunderstanding.

We have reformulated the sentence and it now reads “*Once we identified and partially characterised the components of the arbitrium system present in the Katmira phage, and in order to shed light on the mechanism of action for these receptors, we tried to establish the molecular basis of the AimR^{Kat}-AimP^{Kat} interaction.*”

9. Line 246-24: Structural comparison showed please in supplementary data, to know similar or identical? If difference, please clarify and interpret.

AimR^{SP β} and AimR^{Kat} are dimers, as confirmed by MALS. The structures show that the dimeric conformation of both proteins is similar since dimer interface involves two surfaces, the first one corresponding (or involving) residues from the TPR^{N-ter} subdomain and the second one involving residues from the TPR^{C-ter} subdomain. The interaction surface is highlighted in Supplementary Figure 2. The use of both surfaces generates an almost identical dimer for both structures, as confirmed by the structural superposition shown in Supplementary Figure 3 (upper panel). As there are small differences in the superposition (notice that 760 CA atoms are superposing) we prefer to say “similar” instead of “identical” to avoid confusion. Oppositely, the structures reported by other groups (named in the paper as AimR^{SP β -II}) show a single dimerization surface involving residues from the TPR^{C-ter} subdomain, and consequently the dimeric conformation in these structures is different to that observed in the superposition shown in Supplementary Figure 3 (middle and lower panels). These differences are explained in detail later in the same section.

10. Line 252-255 : The crystals of two of these structures, PDBs 6IPX and 6JGS, show the same space group (P21212) and almost identical unit cell, and were obtained in similar crystallization conditions (Supplementary Fig. 2). Three questions of this sentence: 1. Where to find 6JGS? 2. which crystals crystallization conditions is similar to 6IPX? 3. Which are the same space groups and the same conditions. Please clarify

1. We introduced a typo in this sentence indicating PDB 6JGS instead of 6JG5. We have corrected this mistake.

2-3. 6IPX crystallises in the same conditions described for 6IM4. Both crystals were obtained using a similar mother liquor (20 % PEG 4K, 50 mM magnesium acetate, pH 6.5-7.5), share the same space group (P2₁2₁2) and similar cell constants (a=121, b=214, c=33), despite the fact that 6IPX is the apo form and 6IMP has been crystallised in the presence of the peptide. This is also the case for 5XYB/5Y24 or 6JG5/6JG9. Notice that 6JG5/6JG9 is a reoptimised crystal and a new data set of 5XYB/5Y24 has been carried out by the same authors. For these crystals the apo/peptide pairs also share space group (P2₁2₁2) and similar cell constants (a=115-121, b=214-220, c=33.6) and crystallisation conditions (9-11%PEG 8K (9), NaBr/NaCl, pH 6.0-6.6). The data is summarised in the table below and in Supplementary Figure 2. It can therefore be concluded that all of these correspond to the same crystal forms regardless the presence or absence of peptide. We have reformulated the sentence to make it clearer for the readers and it now reads “*Besides this, 6IPX and 6JG5 structures were solved from crystals showing the same*

space group ($P2_12_12$), almost identical unit cells and similar crystallisation conditions (Supplementary Fig. 2), indicating that they correspond to the same crystal form.”

Structure	Conditions	Space Group	Unit cell (Å)
6IPX (apo)	Hepes (pH 6.5-7.5), PEG 4K, 50 mM magnesium acetate	$P 2_1 2_1 2$	a=120.5 b=214.7 c=33.7
6IM4	Hepes (pH 6.5-7.5), 20% PEG 4K, 50 mM magnesium acetate.	$P2_12_12$	a=121.6 b=214.2 c=33.9
6JG5 (apo)	Sodium cacodylate (pH 6.6), 11% PEG 8K, NaCl	$P 2_1 2_1 2$	a=115.3 b=219.4 c=33.5
6JG9 (peptide)	Sodium cacodylate (pH 6.1), 9% PEG 8K, NaBr, DTT	$P 2_1 2_1 2$	a=121.0 b=214.0 c=33.6
5XYB (apo)	Sodium cacodylate (pH 6.0), 10 % PEG 8K, NaCl, DTT	$P 2 2_1 2_1$ (Should be indexed as $P2_12_12$)	a=33.6 b=114.8 c=220.4 a=114.8 b=220.4 c=33.6
5Y24 (peptide)	Sodium cacodylate (pH 6.1), 9 % PEG 8K, NaBr, DTT	$P 2 2_1 2_1$ (Should be indexed as $P2_12_12$)	a=33.6 b=119.6 c=214.4 a=119.6 b=214.4 c=33.6

11. Line 256: 6JGS. Correct

Corrected.

12. Line 265-273: There are not sufficient evidences to support this explanation. I went through the reference paper, and present my understanding below: AimR_{Kat} and AimR_{SPb-I} apo structures are similar to structure of apo phAimR in close state, but both of them show conformational changes when bound to cognate ligands. So here the current structure of AimR_{Kat} is probably one state of the serially dynamical conformations, and as well as AimR_{SPb-I} and AimR_{SPb-II}.

We fully agree with the Reviewer that AimR receptors (SP β , Katmira and phi3T) in their apo states have a wide conformational dynamic range as was confirmed by our previous structures of apo AimR^{SP β} (doi: [10.1016/j.molcel.2019.01.025](https://doi.org/10.1016/j.molcel.2019.01.025)). This plasticity of the apo AimRs is allowed by the N-terminal slipping dimerization surface in conjunction with a fix C-terminal dimerization surface generated by the C-terminal capping helices. However, in the AimR^{SP β -II} structures, the N-terminal interaction surface is completely loose due to the presence of a C-terminal tag that interfere in the dimerization. Notice that the use of N-terminal and C-terminal dimerization elements are observed in the SP β , Katmira and

phi3T apo AimRs, thus AimR^{SPβ-II} structures do not represent one state of the serially dynamical but rather a conformational state induced by the presence of the C-terminal tag. In fact, binding of the peptide, which controls the receptor activity, inexplicably does not lead to any conformational change in the AimR^{SPβ-II} structures while it does in AimR^{SPβ-I}, AimR^{Phi} and the AimR^{Kat} structure reported in this manuscript. For these reasons, we consider that there is evidence to differentiate between 2 types of conformations, those we call AimR^{SPβ-I}, which, as the reviewer indicates present plasticity, and AimR^{SPβ-II}, which does not present this biological plasticity and is induced by the presence of the C-terminal tag, as later is confirmed in the manuscript.

13. Line 359: The C-terminal His tag disturbs dimer interface in AimR. No sufficient evidences prove the conclusion. Explaining why it could rescue *in vivo* using the C-terminal His-tag in previous report.

To generate evidence of C-terminal His tag disturbance of dimer interface beyond the structural ones and to addresses the questions made by the reviewer we decided to carry out two additional analysis using AimR^{SPβ} with and without C-terminal His tag. First, we carried out thermal stability assays showing that the presence of the C-terminal His-tag induces a strong decrease (9°C) of the melting temperature, supporting a weaker and less stable dimer due to the loss of a large part (N-terminal) of the dimerization. Second, we measured AimR^{SPβ}-DNA operator affinity by BLI. The results obtained, which are explained in the answer to the question 5, confirms that the presence of C-terminal His-tag reduces one order of magnitude the affinity of AimR^{SPβ} for the DNA operator. This reduction is due only to changes in the K_{on} , supporting that the C-terminal His-tag hinders the conformational changes required for DNA binding. Once the C-terminal tag is expelled from the N-terminal surface, as is observed in PDB 6JG8, the receptor can acquire de DNA competent conformation identical to the observed in the structure of AimR^{SPβ-I} in complex with DNA (PDB 6HP7) as is supported for the identical K_{off} for AimR with and without His-tag in the BLI assays. The structure of AimR^{Kat} in complex with DNA included in this version of the reviewed manuscript also confirm that the DNA binding conformation, that it is almost identical to AimR^{SPβ}-DNA (RMSD ~2.1 Å) requires the use of two dimerization areas and, consequently, the projection of the C-terminal tail to the solvent to eliminate the contacts that prevents the use of the N-terminal dimerization site. These observations also explain why the heterologous expression of the receptor with the C-terminal tail can rescue *in vivo* a deletional mutant, as the presence of DNA stabilizes the biological conformation by forcing the movement of the C-terminal tail. Although this fact has an energetic cost, as is reflected in a decrease of the DNA association constant, it is compensated by higher levels of protein that is expressed in the *in vivo* complementation assays.

We hope that these new experimental data, which have been integrated into the manuscript and collected in Table I, Figures 5 and S8, are sufficient evidence to support the disturbed role of the C-terminal His-tail.

14. It should highlight that the structure of phAimRY341A/E371A-AimP in monomer, not wild type, was a double mutant variant, the presentation of the current manuscript would lead to misunderstanding.

We completely agree with the reviewer. Indeed, these mutations seem to be the cause of the monomeric character of AimR^{Phi} in the presence of the peptide as shown in a recent publication (doi: 10.3390/biom11091321) that revisited its mechanism of action using different technical approaches and confirming that AimR^{Phi} is also a dimer in both apo and peptide-bound forms. We have made changes throughout the manuscript to reflect these data, acknowledging the new publication and confirming a similar mechanism of action for all these AimRs.

15. In the manuscript, several paragraphs discussed the effect of the C-terminal tag of AimRspBeta, but never talk about the dynamics of AimRph, and never discuss about the effect of the C-terminal tag of AimRph. The author even considerate that SAIRGA could disrupt the dimeric state of AimRph. Structures of these receptors or bound to ligands are probably different snapshots, which have been indicated by gel-filtration assays. The authors should clarify these different structures from the same spBeta phage group, why are same, why not.

It should be noted that the structures of the AimR^{Phi} mutant were made with a protein that lacked a C-terminal tag, since the GST-tag used for expression and purification was cut before crystallisation so not to disturb the dimer interface. As indicated in the previous answer, the peptide-induced monomerization observed AimR^{Ph} is only attributable to mutations located in the C-terminal dimerization region as recently confirmed by a new publication (doi: 10.3390/biom11091321). This publication shows that the peptide does not induce AimR monomerization of wild-type AimR^{Phi}. Unfortunately, there is no structural data with wild-type AimR^{Phi} but the AimR^{Phi} mutant structures show that the conformational changes for each AimR monomer are similar to those observed in AimR^{SPβ}, since the peptide induces a locked conformation where the TPR^{N-ter} and TPR^{C-ter} subdomains approaches. In fact, the RMSDs for the superposition of AimR^{Phi}, AimR^{SPβ}-I and AimR^{Kat} monomers are very low, confirming a similar mechanism of action all these AimR receptors. We remodelled the text to incorporate the new evidence.

16. One more structures should be solved of the AimR superfamily, the spBeta subfamily is the small part. In particularly, the identity of these protein would be better if lower 30%. Thus, it will provide insights into the differences between AimRKat and AimRSpB and AimpPh, and other receptors.

Although we considered that solving the structure of an unrelated subfamily of AimR was beyond the scope of this manuscript, we took the suggestion made by the Reviewer seriously and have solved the structure of of AimR^{Kat} in complex with its target DNA, which is presented in the new section "*Crystal structure of AimR^{Kat}-DNA complex confirms the mechanism of action*" and Fig 5. This additional structure confirms our molecular mechanism of action for AimRs in clade II.

17. In Fig. 1b The peptide sequences of AimpSpB and AimpKat are wrong.

Corrected.

18. In Fig. 1c the DNA Sequence name should be AimXSpB or Kat

We changed it to *aimX*-box^{SPβ} and *aimX*-box^{Kat} to clarify the reference to the box located upstream of *aimX*. Thank you for the correction as it is now clearer for the reader.

19. In Fig. 1d, the DNA operator of phi3T indicate have a bit interaction with AimRkat, clarify please.

As the Reviewer notices there is a little shadow in the EMSA migrating slightly slower than the other AimR-DNA complexes. However, the unbound DNA is not disappearing. We believe that it must be due to non-specific binding since the BLI analyses now carried out have shown no binding of AimR^{Kat} to *aimX*-box^{Phi} (Table I).

20. In Fig. 1e or legend, the Kd has to be indicated.

The calculated Kd has been added to both Fig. 1e and legend.

21. In Fig. 2a. Error bar is approximate zero, interpret please.

We would like to thank the Reviewer for pointing this out. We have included several more replicates to validate these data and included the individual data points with more visible error bars. Further, we have corrected the statistics used to interpret the comparisons of these data.

22. In Fig. 2b, the parallel experiments in curves as supplementary data or main figs are needed.

As the Reviewer notices the explanation as to how this experiment was performed was lacking some clarity. We have previously shown that the complementation of $\Delta aimR^{SP\beta}$ produces sharper plaques but does not change the titer of the mutant (Brady et al., 2021). For this experiment, a dilution from a phage SPβ wt or $\Delta aimR$ lysate was titered into the different recipient strains to obtain around 200 pfu. Hence, the titer is the same for all the recipients used, but the plaque morphologies observed are different depending on the recipient strain. The Figure 2b legend has been changed to include a more complete explanation and it now reads “*Lysates from phage SPβ wt and $\Delta aimR$ were used to titer into B. subtilis 168 $\Delta 6$ and $\Delta 6$ with the Pspank cloned gene of aimR SPβ and Katmira as the recipient strain. A dilution of these lysates was performed to visualise around 200 pfu. When indicated 5 μM of peptide AimP^{SPβ} or AimP^{Kat} was added before plating. The resulting plaque morphologies were photographed.*”

23. Fig. 5b, label or legends represent clearly. Through all text, authors highlight that the regulation mechanism of AimRKat is most similar to AimRSpB, and their structures should be identical. To support this point, structure of AimRKat-AimXkat is requested, then overall structural comparisons between apo-AimRSpB and AimRKat or bound to DNA ligands are needed.

We have solved the crystal structure of AimR^{Kat} in complex with its target DNA (*aimX*-box^{Kat}). It is widely described in the section “*Crystal structure of Aim^{Kat}-DNA complex confirms the mechanism of action*”, lines 372-403. As the Reviewer suggests, it helps to support the mechanism proposed and reinforces the manuscript conclusions.

24. In Fig 6. C, the result shows AimpSpB could induce DNA release as well as Aimpkat, it essentially needs in vivo experiments (knock-out of AimRSpB or Aimpkat of corresponding phage stains, respectively, then perform complement experiments using cross receptors) to further confirm this conclusion, moreover, the measurement of binding affinity for AimRKAT-N273A and AimpSpB is also essentially needed. More experiments and discussion are needed to substantiate the current point.

As suggested, the binding affinity of AimR^{Kat}-N273A for AimP^{Kat} (GIVRGA) and AimP^{SPβ} (GMPRGA) were calculated by ITC. Although the affinity for AimR^{Kat}-N273A binding to AimP^{SPβ} is 100 times lower than that of AimR^{Kat}-AimP^{Kat}, it is in the order of micromolar and has been achieved with a single substitution. While analysing the results, we realised that the curves obtained were biphasic and the adjustment fitted better to a multiple-binding site model. We believe this makes sense, as binding of the peptide to one monomer would induce a conformational change that alters the binding site for the other monomer of the dimer. In the wt AimR, the affinities of both sites are so similar (4.1 and 5.7 nM) that was difficult to interpret initially. We measured the affinities for AimR^{Kat}-AimP^{Kat} again with smaller volume injections and we could see the biphasic curve. In fact, we find this finding very interesting and will continue studying it. We thank the Reviewer for bringing this point to our attention.

25. More dynamical experiments could further exemplify the conclusion of this manuscript.

As the Reviewer request additional tests have been performed (BLI, thermal shift, structure determination) and the conclusions of the manuscript have been reinforced.

26. In this work, the author characterized the structure of AimRKat, which is almost identical to AimRSpB, However, many structures have been reported before (References 6-9), the author has cited them, it will be important to objectively point out the difference and similarity based on in vivo, in vitro functional experiments and more structures among the findings in current work and the previous reports to avoid overstating and misunderstanding.

As the Reviewer indicates there is extensive structural information for AimP^{SPβ} but, as is shown in this manuscript and summarised in Supplementary Figure 2, many of these models correspond to basically the same structure solved and published by different authors. Throughout the manuscript, we have tried to objectively show the differences between this and other models by moving point by point and comparing them in their different states (apo, peptide-bound and DNA-bound) as shown in Supplementary Figures 3, 6, 8 and 9 among others. In this way we have provided readers with all available structural information and its comparison to avoid overstating and misunderstandings.

Reviewer #2 (Remarks to the Author):

Professor Alberto Marina and the co-authors determined the structure of apo AimRkat and the complex with the peptide (GIVRGA) in this manuscript. However, they and other

groups solved the structure of AimRbeta-AimP and AimRPhi-AimP almost simultaneously in 2019. The structural basis of how AimR recognized AimP and how AimP prevented AimR binds to AimR are clear. Although the author noted that a little difference between their results and they further proved their hypothesis in this study that the AimRKat with a chimeric trait, which is further supports their point. But I believe the novelty is not enough for the publish in Nature Communications. I suggest that they can consider for transferring it to Communication Biology.

Since its discovery, the arbitrium system has gained a lot of interest. This is because it represents a fascinating mechanism of communication between phages. Importantly, recent papers in the area have demonstrated that this system not only controls phage infection, but also prophage induction (DOI: 10.1038/s41564-021-01008-5; DOI: 10.1016/j.cub.2021.08.073; DOI: 10.1016/j.cub.2021.08.072). Therefore, while the interest of the system is clear, we do not really know how it really works, with two different (and incompatible) mechanisms of action proposed. In this manuscript we solve this mystery and clearly establish the molecular basis of how the system works. We humbly think that this work deserves publication in this journal.

I also have two questions:

1. I am interested that if does it impact the ability of AimR binds to AimX if the 6XHis tag in the C terminus? Because the structure of AimR-AimX(PDB: 6GJ8) was also solved. Normally, if the transcription factors are not in the right condition, it can not bind the promoter, so please compare the affinity between AimR-6XHis and AimR alone with AimX.

The reviewer is right, and our new results confirm their hypothesis. The discrepancies between the two models that were proposed to explain how the arbitrium system works were because the presence of this C-terminal His-tags in some of the proteins. Our BLI assays show that the presence of the tail decreases the affinity for the operator by one order of magnitude, with the decrease falling almost exclusively on K_{on} and not on K_{off} , which agrees with the structural data (see answers to Reviewer #1). Nevertheless, the K_D is high enough to allow DNA binding. Moreover, thermal stability assays show that the presence of the C-terminal His-tag induces a strong decrease (9° C) of the melting temperature, supporting a weaker and less stable dimer due to the loss of a large part (N-terminal) of the dimerization surface as shown by the structural data.

2. A blank space should be between the number and their units such as 150 mM, there are too many little errors in this manuscript, please check it.

We have carefully reviewed the manuscript and hope to have corrected all typos.

Reviewer comments, second round of review

Reviewer #1 (Remarks to the Author):

This revised manuscript is much improved compared to the previous version that had many serious problems. In the revised version, the authors have done additional essential experiments to fix these problems. Now, I think the current study almost could be fine for publication in NC, but there are still issues that the authors should address.

1. the binding affinity: why here showed two binding site? how to explain and these are different with other group reports, it should clarify and provide solid experimental data. I am afraid of that this result is not solid
2. the authors still stucked to that the C-terminal 6*His tag effected the confirmation, could they provide convincing evidence to prove, for example, it could be used the construct with C-terminal 6*His tag or without expressed in vivo to check the phenotype.
3. this new structure, with more than 40% similarity, is a homolog of the previously reported Sp/Ph proteins, it should be serious to make the conclusion about the crosstalk, IMHO, it is not perfect.

REVIEWER COMMENTS:

Reviewer #1 (Remarks to the Author):

This revised manuscript is much improved compared to the previous version that had many serious problems. In the revised version, the authors have done additional essential experiments to fix these problems. Now, I think the current study almost could be fine for publication in NC, but there are still issues that the authors should address.

We thank the reviewer for their initial and current comments, which have undoubtedly helped to significantly increase the quality of the manuscript.

1. the binding affinity: why here showed two binding site? how to explain and these are different with other group reports, it should clarify and provide solid experimental data. I am afraid of that this result is not solid

We apologise for not having properly explained the binding mechanism of AimP peptide to AimR. AimR is a dimer and each monomer has only one peptide binding site. After the initial comments of the Reviewer, we measured by ITC (Isothermal Titration Calorimetry) the AimP binding affinity to the AimR^{Kat}-N273A mutant and we realised that the thermogram showed a biphasic shape and should be better adjusted to two binding events (two-binding sites according to software terminology). This does not mean that each AimR monomer has two AimP binding sites but rather that there is some allosteric effect between the unique sites of each monomer within the AimR dimer. This makes sense since the binding of the peptide to one of the monomers causes important conformational changes (as the structures reported in the manuscript confirm) that can alter the binding site in the second monomer. We did not notice this allosteric effect in our first version of the manuscript as it is minimal for the wild-type protein, with both subunits of the dimer showing almost identical affinity for the peptide (4.1 and 5.2 nM), so the thermogram could be adjusted to a model with identical binding affinities in each monomer. We think that this is why other groups have not mentioned this effect in their manuscripts. When the affinities are lower (as in the case of the mutant) and the saturation of the thermogram is slower is easier to recognise this biphasic behavior that should be adjusted to a two-binding site model. After mutant affinity characterisation we repeated the AimR^{Kat} wild-type titration adjusting peptide concentration to avoid rapid saturation and we observed the biphasic curve, as can be seen in Figure 1e in the new version.

The reviewer's comment made us realise that the nomenclature and description of this allosteric effect between peptide binding sites was not the most accurate. To avoid confusion, we have named the K_D values as K_{D1} and K_{D2} for monomer 1 and monomer 2 instead of site1 and site2 to reinforce the idea that it is referring to the same binding site but one in each monomer of the functional AimR dimer.

We changed lines 205-211 and it now reads: "*The experiment showed a biphasic thermogram, suggesting an allosteric effect between the two AimP binding sites, one in each AimR^{Kat} monomer, within the dimer. However, the peptide GIVRGA binds to the AimR^{Kat} dimer with a similar high affinity at both peptide sites (K_D values, K_{D1} 4.1 ± 3.2 nM and K_{D2} 5.7 ± 2.1 nM for monomer 1 and monomer 2, respectively), supporting a*

weak but existing cooperativity between the two AimP-binding sites on the dimeric AimR receptor (Fig1e)”

2. the authors still stuck to that the C-terminal 6*His tag effected the confirmation, could they provide convincing evidence to prove, for example, it could be used the construct with C-terminal 6*His tag or without expressed *in vivo* to check the phenotype.

Our structural results with two different AimRs show that the presence of His-tag induces a conformational change that is confirmed *in vitro* by a decrease in affinity for the DNA binding site, although this decrease is moderate and the AimR with His-tag still maintains DNA affinity on nanomolar range. The structural data reported also explains this phenotype, showing that the presence of the His-tag does not prevent binding to the DNA, since as observed in the structure reported by Guan and collaborators (PDB 6JG8; doi.org/10.1038/s41421-019-0101-2), to bind to the DNA, the AimR His-tag is displaced from the dimerisation surface projecting into the solvent, thus allowing AimR to adopt the DNA-binding competent conformation. In addition, the thermal shift assays for AimR^{SP β} also confirms that the C-terminal His-tag induces a decrease of more than 9 degrees in the AimR^{SP β} Tm versus the protein without the tag, supporting the impact of the His-tail in the AimR conformation that makes the dimer less compact and stable as the structures shown. In summary, we provide multiple evidence showing that the presence of the tail affects the conformations adopted by the protein but does not disable it from performing its function. We have acknowledged this in the manuscript in lines 527-536 (new version numbering) that reads:

*“This fact is reflected in the DNA binding analysis since the comparison of the k_{on} and k_{off} constants for both AimR^{SP β} proteins shows that while the k_{off} is identical, the k_{on} is five times lower in the case of AimR^{SP β} -II (Table 1), confirming that the presence of the C-terminal tag hampers AimR from acquiring the competent conformation before DNA binding. However, once acquired, for which this tag must be expelled from the dimerisation interface as shown in the AimR structure (Fig. 6 and Supplementary Fig. 8), the binding is not affected. In addition, this DNA-induced conformational change explains why the heterologous expression of the receptor with the C-terminal tail can rescue *in vivo* a deletion mutant^{10,11} although its affinity for the target DNA is significantly lower”.*

We consider these results to be consistent, providing a clear mechanism of action on how AimP blocks AimR function. By contrast, we would like to mention here that in the other proposed model, the mechanism of action is merely speculative, since the authors could not see any conformational change in AimR after binding to AimP. Moreover, it is important to remember that the conformational changes induced by the peptide is the characteristic mechanism of action proposed for the regulatory peptides in the RRNPP quorum sensing receptor family. Therefore, our proposed model of action perfectly fits with what has been previously published for these types of receptors.

In any case, following the reviewer's indications and to evaluate this effect *in vivo*, we cloned *aimR*^{SP β} and *aimR*^{Kat} genes with and without C-terminal His-tag into the *amyE* integration vector pDR110 under the control of the IPTG inducible promoter *P_{spank}* and we carried out complementation assays *in vivo* using a $\Delta aimR$ phages. The growth dynamics after induction or infection showed slight differences between the version with and without His-tag but these differences were highly variable and poorly reproducible,

making our results not consistent enough to be published. We believe that this variability is partly due to the fact that by using a strong promoter, AimR is produced at high and variable levels, masking the modest difference in affinity induced by His-tag. We also believe some of the variability observed in our *in vivo* experiment was due to the complex lifecycle of our model organism *B. subtilis*. Our experiments required the lysogenic strains to be in exponential growth, however, it is likely that part of the population is undergoing an alternative lifecycle (e.g., sporulation, biofilm production) and with this we see some variability *in vivo* that made it difficult to identify the differences between complementation with either the wt AimR or AimR-His. We honestly believe that our structural and *in vitro* results are sufficiently strong to support the conclusions presented in the manuscript and to carry out these tests would delay the publication of the manuscript too much, in addition to the workload it would entail.

3. this new structure, with more than 40% similarity, is a homolog of the previously reported Sp/Ph proteins, it should be serious to make the conclusion about the crosstalk, IMHO, it is not perfect.

We apologise if the manuscript gives the appearance that we concluded an existence of crosstalk among the arbitrium systems, which was not our intention since we have no experimental data to support it.

We believe that the wording of the manuscript at no time intended to give that message. The revised version of the manuscript only mentions crosstalk 3 times. Once in the results section and twice in the discussion.

In the results section we only indicate that the differences in the sequence of the TPR domains (>50 %), responsible for peptide recognition, must account for avoiding crosstalk between AimRs.

While in the discussion section, following our mutagenesis results where we can vary the affinity for the regulatory peptide with a single mutation between two related AimRs (SP β and Katmira) we speculate on the possibility of possible crosstalk between related phages. We consider this to be only a personal vision and a proposal for future work, as we want to make clear in the text "*Our vision is that related phages may not only present cross regulation but also crosstalk, showing the arbitrium system to be involved in complex phage-phage interactions. Our proposition is that these interactions will confer social behaviors to arbitrium-carrying phages.*" and was made in the discussion section.

To make it clear that this is only a working hypothesis to be confirmed, we have rephrased this section and now reads: "*Our vision, which is entirely speculative, is that related phages may not only present cross regulation but also crosstalk, showing the arbitrium system to be involved in complex phage-phage interactions. The confirmation of this hypothesis would open the interesting possibility that this quorum sensing mechanism could confer social behaviours to arbitrium-carrying phages.*"

Reviewer comments, third round of review

Reviewer #1 (Remarks to the Author):

The authors addressed the mentioned questions and concerns, now, it should be suitable for publication in Nat Commun.